# The Quantum Regularization of Singular Black-Hole Solutions in Covariant Quantum Gravity

**DOI:** 10.3390/e23030370

**Published:** 2021-03-20

**Authors:** Massimo Tessarotto, Claudio Cremaschini

**Affiliations:** 1Department of Mathematics and Geosciences, University of Trieste, Via Valerio 12, 34127 Trieste, Italy; 2Research Center for Theoretical Physics and Astrophysics, Institute of Physics, Silesian University in Opava, Bezručovo nám.13, CZ-74601 Opava, Czech Republic; claudiocremaschini@gmail.com

**Keywords:** quantum gravity, space-time singular solutions, quantum regularization

## Abstract

An excruciating issue that arises in mathematical, theoretical and astro-physics concerns the possibility of regularizing classical singular black hole solutions of general relativity by means of quantum theory. The problem is posed here in the context of a manifestly covariant approach to quantum gravity. Provided a non-vanishing quantum cosmological constant is present, here it is proved how a regular background space-time metric tensor can be obtained starting from a singular one. This is obtained by constructing suitable scale-transformed and conformal solutions for the metric tensor in which the conformal scale form factor is determined uniquely by the quantum Hamilton equations underlying the quantum gravitational field dynamics.

## 1. Introduction

The discovery of the characteristic central singularity that may characterize black holes (BH) is due to the genius of Karl Schwarzschild who in 1916 pointed out his famous exact solution [1] of Albert Einstein’s namesake field equations, soon brilliantly followed by Hans Reissner [2] and Gunnar Nordstrôm [3] who generalized it to the case of a charged BH, namely a Schwarzschild-type BH carrying a total net charge *Q*. This motivated much of the subsequent spur of related investigations. Nevertheless it was only in 1963 that Roy Kerr, extending the Schwarzschild solution, discovered the exact solution for a vacuum rotating object in general relativity [4]. Notably, however, the interpretation of a BH as a region of space from which nothing can escape, although based to an earlier theoretical prediction formulated in 1939 by Oppenheimer and Snyder [5], is usually attributed to a paper published many years later in 1958 by David Finkelstein [6]. Finally, the term “black hole” itself was coined only in 1967 by John Wheeler [7] (before that the names “singularity” [8], “frozen star” [9] or “collapsed star” [10] were commonly used to refer to such objects). Meanwhile, investigations pointed out that BHs are a frequent occurrence in classical general relativity (GR [11,12]), including among others the Kottler–Schwartzchild–deSitter, Reissner–Nordstrom and the Freeman-Lemaitre-Robertson-Walker(FLRW)–Schwarzschild–deSitter cases (the first two being stationary, namely independent of coordinate time when expressed in suitable native coordinates [13]).

In the following we restrict our analysis to the case of classical BH singularities. A semantic clarification must be given concerning the behavior of the solution referred to here as “singular”. In fact, by singularity of the metric field tensor, we mean the singularity that occurs in the center of the BH (i.e., the origin of the coordinate system) and which cannot be eliminated by means of suitable changes of GR-frame (coordinate system) to be realized only by means of local point transformations (LPT). The latter ones are coordinate diffeomorphisms of the type
(1)$$r\to r^{\prime }=r^{\prime }\left(r\right),$$
forming the so-called LPT-group, which leaves unchanged the differential manifold of space-time $\left\{Q^{4},\hat{g}\left(r\right)\right\}$, hereon for definiteness identified with a time-oriented 4−dimensional Riemannian space-time with signature $\left\{1,-1,-1,-1\right\}$. Here, $\hat{g}\left(r\right)$ denotes the associated classical metric field tensor solution of the Einstein field equations (EFE), parametrized with respect to a coordinate system $r\equiv \left\{r^{\mu }\right\}$ and defined via its covariant and countervariant coordinate representations $\left\{{\hat{g}}_{\mu \nu }\left(r\right)\right\}$ and $\left\{{\hat{g}}^{\mu \nu }\left(r\right)\right\}$. More precisely, adopting a coordinate system in which the geometric center of the BH identifies the origin of spatial coordinates, the singularity we are referring to here is actually that which characterizes the covariant components ${\hat{g}}_{\mu \nu }\left(r\right)$ and in particular its time–time component ${\hat{g}}_{00}\left(r\right)$ which exhibits a divergent behavior when approaching the origin r=0. This occurs when at least one of these components diverges (i.e., it is not locally defined, together with the corresponding countervariant components ${\hat{g}}^{\mu \nu }\left(r\right)$). However, there is here another possible related issue which arises. In fact the singularity affects also the prescription of the Riemann distance, namely
(2)$$ds^{2}={\hat{g}}_{\mu \nu }\left(r\right)dr^{\mu }dr^{\nu }.$$

As a consequence, when some of the components ${\hat{g}}_{\mu \nu }\left(r\right)$ locally diverge it follows that certain contributions of the infinitesimal displacement tensor $dr^{\mu }$ must vanish identically in order to warrant the regular character of ds, a requirement that may be in possible contradiction with other fundamental physical requirements, such as the Heisenberg uncertainty principle.

Nowadays it is generally acknowledged that space-time singularities, particularly BH ones, actually play an essential role in GR, due to their widespread nature. On the other hand, the very existence of such singularities represents a crucial conceptual issue, possibly related to the limits of validity of GR itself, since these singularities cannot be resolved/cured in the framework of classical GR or by recurring to higher-order curvature and non-local models of classical gravity. On the contrary, the prevailing opinion is that such singularities should be regarded as signatures of possible quantum effects that occur in the presence of intense gravitational fields [14,15,16]. This is indeed one of the main motivations that lies behind the investigation of strong field regimes of gravity through the direct observation and detection of gravitational waves and BHs. Thus, properly understanding the role of quantum gravity becomes increasingly urgent and meaningful.

The conjecture is that QG, realized by means of a suitable quantum theory of the gravitational field, should allow the achievement of smoothly continuous and everywhere-regular geometric representations of space-time. To state it more precisely:The regularization should be carried out by means of suitable quantum-based modifications of EFE capable of smoothing out all classical BH singularities.Such a regularization should have a universal character, i.e., it should hold for arbitrary singular BH solutions.The said regularization should not require the introduction of “ad hoc” extra classical or quantum fields.

The same theory of QG, in other words, should be capable of resolving the mathematical BH-singularities arising in classical GR, thus warranting the regularity of the background metric tensor. Needless to say, however, the goal should be reached without introducing any unwanted pathological behavior, such as:discontinuities and singularities associated with discrete quantum theories, which possibly violate, besides continuity, the principle of general covariance and the differential manifold structure of space-time;the occurrence of absolute minimum lengths, a feature that by itself implies breaking the principle of general covariance;intrinsically frame-dependent theories, such as ADM quantum theory, violating some of the fundamental symmetries characteristic of EFE, i.e., the properties of manifest covariance and gauge invariance.

Possible relevant applications include both the description of the structure and dynamics of the universe in the framework of cosmology, as well as the prediction of quantum phenomena arising in GR scenarios associated with black holes and event horizons (EH).

Nevertheless, the identification of the relevant quantum phenomenology depends very much on the precise choice of the model of quantum gravity to be adopted. Therefore, the choice of the quantum gravity model becomes an issue by itself. In this regard, one of the practical obstacles to most attempts to quantization of classical gravity in GR is undoubtedly the vast complexity of some of these theories. A feature that makes quantitative comparisons or even simple logical rational deductions based on such theories is practically impossible. One such case is the notorious issue about the (possible) quantum regularization of BH singularities.

However, the manifestly covariant nature of EFE, as well of all relativistic classical and quantum theories also outside GR, suggests a possible censorship on the class of admissible quantum theories.

In fact, just like classical theories, also quantum theories and in particular QG should satisfy, at a certain level, the so-called manifest covariance principle, requiring their frame-independent character, namely their tensor property with respect to the group of local point transformations (Equation 1). Such a property, however, necessarily demands the adoption of a so-called “background” space-time viewpoint. In other words, a “background” space-time picture should be adopted, where space-time should be represented by a differential manifold $\left\{Q^{4},\hat{g}\left(r\right)\right\}$, with $Q^{4}\subseteq R^{4}$ being a 4−dimensional time-oriented (“background”) Riemann space-time and its metric field tensor
(3)$$\hat{g}\left(r\right)\equiv \left\{{\hat{g}}_{\mu \nu }\left(r\right)\right\}\equiv \left\{{\hat{g}}^{\mu \nu }\left(r\right)\right\}$$
to be considered prescribed (i.e., once the coordinates $r\equiv \left\{r^{\mu }\right\}$ are defined). Such a tensor field is referred to as “background” metric tensor.

In this regard, a serious obstacle (which most of such theories exhibit) occurs already at the classical level. This is the issue of the rigorous connection between QG and the classical Einstein field equations, which should be suitably recovered in the context of QG. As a consequence, since EFE is manifestly covariant, i.e., it is set in manifest 4−tensor form with respect to the LPT-group (Equation 1), an obvious requirement to be set on QG is that it should exhibit the same property of manifest covariance.

Despite major theoretical developments achieved in the past, a theory fulfilling the same principles has remained until very recently largely unsolved. The fundamental reason is that a corresponding manifestly covariant, and possibly constraint-free, classical Hamiltonian theory of GR is actually required for the completion of such a task, a feature that is missing in previous literature. For this reason in this paper the so-called manifestly covariant approach to QG (CQG-theory) recently developed in Refs. [17,18,19,20,21,22,23,24,25,26,27,28,29] will be adopted.

The reason why, ultimately, CQG-theory should be considered as a possible candidate adequate for the task is that it is based on a manifestly covariant and truly classical Hamiltonian structure for EFE. In other words, denoting $r\equiv \left\{r^{\mu }\right\}$ a generic 4−position, i.e., a point of the set $Q^{4}$, this means that such a Hamiltonian structure should be necessarily represented by a set of the type $\left\{x_{R}\left(r\right),H_{R}\right\}$, with $x_{R}\left(r\right)\equiv \left\{g_{\mu \nu },\pi^{\mu \nu }\right\}$ and $H_{R}$ denoting respectively a suitable classical canonical state with $g_{\mu \nu }$ and $\pi^{\mu \nu }$ classical variational tensor fields and an appropriate classical Hamiltonian density. We stress that all these quantities, i.e., $g_{\mu \nu }$, $\pi^{\mu \nu }$ and $H_{R},$ are identified with suitable 4−tensor fields with respect to the aforementioned background metric field tensor $\hat{g}\left(r\right)$ (Equation 3). However, in order for such Hamiltonian structure to exist, both the said canonical state and the Hamiltonian density must depend parametrically on a suitably-prescribed 4−scalar dynamical parameter *s*, denoted as proper time. As a consequence, this means that both the canonical state $x_{R}\left(r\right)$, the background field $\hat{g}\left(r\right)$ and the Hamiltonian density $H_{R}\left(x_{R}\left(r\right),r\right)$ must be considered as suitably parametrized in terms of *s*. This is achieved by setting, in particular, r=r(s), where *s* is the arc length (proper-time) prescribed along a geodesic curve $C_{\left(r_{o},r_{1}\right)}$ belonging to a prescribed family of geodesics $\left\{C_{\left(r_{o},r_{1}\right)}\right\}$, which is defined with respect to the background space-time $\left\{Q^{4},\hat{g}\left(r\right)\right\}$. For the appropriate definitions, we refer the interested reader to Ref. [29], where a precise definition of the proper-time *s* is also provided (together with relevant comparisons with the customary ADM theory). Thus, in the same reference the prescription of a generic geodesic curve of the family, $C_{\left(r_{o},r_{1}\right)}$, emerges naturally in the context of a path-integral variational formulation for the classical Hamiltonian structure $\left\{x_{R}\left(r\right),H_{R}\right\}.$ As a consequence, it follows, in particular, that $C_{\left(r_{o},r_{1}\right)}$ is necessarily identified with a finite length geodetics of the type
(4)$$C_{\left(r_{o},r_{1}\right)}=\left\{\left.r\right|r=r\left(s^{\prime }\right),r_{o}=r\left(s_{o}\right),r_{1}=r\left(s_{1}\right),s^{\prime }\in \left[s_{o},s_{1}\right]\;,r_{o}\in \Sigma_{0}^{3},r_{1}\in \Sigma_{1}^{3}\right\},$$
where $\left[s_{o},s_{1}\right]\equiv I\subset R$ denotes a finite proper-time interval, while $\Sigma_{0}^{3}$ and $\Sigma_{1}^{3}$ are two 3D suitable subsets of $Q^{4}$ to which the initial and final 4−positions $r_{o}=r\left(s_{o}\right)$ and $r_{1}=r\left(s_{1}\right)$ belong.

The corresponding quantum theory, denoted as CQG-theory, is based on the manifestly covariant canonical quantization of the classical Hamiltonian structure $\left\{x_{R},H_{R}\right\}$, whereby classical and quantum Hamiltonian field variables or operators, including continuum coordinates, conjugate momenta and Hamiltonian densities are represented by tensor fields. The involved notion of manifest covariance given here is unambiguously defined only by prescription of the differential-manifold structure of the background space-time on which it is displayed and which is self-consistently achieved by CQG-theory (see related discussion in Ref. [23]). The foundations of CQG-theory lie on the preliminary establishment of a variational formulation of classical GR achieved in the context of a covariant DeDonder–Weyl-type approach to continuum field-Hamiltonian dynamics. As such, CQG-theory is endowed with a number of further unprecedented key features, since: (A) unlike ADM theory, it is based on a truly Hamiltonian structure of GR (see Ref. [28]); (B) it preserves the background metric tensor, which is identified with a classical field tensor; (C) it preserves the probabilistic physical interpretation of quantum mechanics to be applied to the quantum gravitational field; (D) it satisfies the quantum unitarity principle, i.e., the quantum probability is conserved in the absence of gravitational sinks; (E) it is constraint-free, in the sense that the quantum variables are identified with independent tensor fields; (F) it is non-perturbative so that the quantum fluctuations of field variables and momentum operators need not be regarded as asymptotically “small” in some appropriate sense with respect to the background metric tensor; (G) CQG-theory provides the physical interpretation of the cosmological constant as being due to quantum Bohm interactions arising among collisionless gravitons.

In view of these considerations, CQG-theory can be said to realize at the same time both a canonical and a manifestly covariant quantization method, in this way overcoming the limitations of former either canonical or non-canonical, but non-manifestly covariant and non-gauge invariant, literature approaches. In fact, it must be stressed that CQG-theory is conceptually intrinsically different and distinguishes itself from these approaches. This provides a promising and innovative theoretical framework that should be regarded as a plausible route (to QG) in view of the axiomatically self-consistent, perspicuous and mathematically-tractable formalism as well as a number of conceptual new features of CQG-theory that depart it in several ways from previous literature. This conclusion is supported by the theoretical outcomes established so far by CQG-theory and the remarkable number of analytical results, experimentally-testable predictions and even basic conceptual innovations achieved so far in such a framework, which concern, for example, the existence of an invariant discrete-energy spectrum for the quantum gravitational field [20] and the consequent graviton mass estimate, the emergent gravity picture related to the generalized-Lagrangian path representation of CQG-theory [22], the novel quantum-gravity interpretation of the cosmological constant as arising due to the Bohm vacuum graviton self-interaction [23,25,26], the non-unitary generalization of CQG-theory due to graviton sinks/sources [24], the quantum screening effect of the cosmological constant [25], the discovery of the stochastic nature of the deSitter event horizon [26], the validity of generalized Heisenberg inequalities expressed in 4-tensor form [21], particularly in connection with the discovery of the proper time-conjugate canonical momentum Heisenberg inequality and the related new statistical interpretation of the concept of invariant minimal length arising in the context of QG [27].

Based on these outcomes, the main goal of the paper deals with the regularization of space-time singularities and the study of the phenomenon of emergent gravity in the framework of manifestly covariant quantum gravity theory. This means exploring the statistical connection between fluctuating quantum gravitational fields and the classical background metric tensor fixing the geometric properties of space-time. Accordingly, from the physical point of view, the background metric tensor should be effectively interpreted as arising from a statistical average of stochastic fluctuations of the quantum gravitational field whose quantum-wave dynamics is described by generalized Lagrangian path trajectories predicted by CQG-theory (see Ref. [22]). The non-local quantum-gravity interaction is expected to permit the non-perturbative mathematical resolution of classical singularities and their physical characterization, suggesting physically-detectable imprints of quantum processes occurring in these contexts. The goal is therefore to address in such a framework the problem of regularization of classical BH solutions. In this regard, an open question concerns what should be the expected characteristic features of such gravitational fields, with particular reference to the following issues:*Preliminary issue #1:* Whether and eventually how quantum gravity models, and specifically CQG-theory, can cure all BH singularities, giving rise to a suitable quantum-modified background metric field tensor (MFT).*Preliminary issue #2:* What is the possible role of the cosmological constant and how its quantum and therefore ubiquitous character could actually be significant for the regularization of singular space-time solutions.*Preliminary issue #3:* What are (if any) the possible large-scale effects produced by the local quantum modifications of MFT.*Preliminary issue #4:* Whether there is a possible connection between the occurrence/prediction of asymptotic/local inflationary regimes, i.e., which are characterized by high values of the cosmological constant, and the expected phenomenon of BH-singularities-quenching.

The present approach is based on the construction of an appropriate conformal-like solution of the quantum Hamilton equations holding in CQG-theory for the quantum gravitational field. Such a solution is shown to be regular in the presence of a Kottler (Schwarzschild–deSitter) background singular BH space-time, as well for the Reissner–Nordstrom–deSitter and the FLRW–Schwarzschild–deSitter space-times. As we intend to show below, in principle arbitrary singular BH solutions can be regularized in the same way.

## 2. CQG-Quantum Hamilton Equations

A characteristic feature of CQG-theory [20] is that the quantum-wave function should be of the form $\psi \left(s\right)\equiv \psi (g,\hat{g}\left(s\right),s,r\left(s\right))$, namely depending simultaneously on the continuous Lagrangian coordinates represented by the variational symmetric field tensor $g\equiv \left\{g_{\mu \nu }\right\}$ which spans the 10-dimensional configuration space $U_{g}\subseteq R^{10}$, and on the background field tensor $\hat{g}\equiv \left\{{\hat{g}}_{\mu \nu }\right\}$, which for definiteness is assumed here of the form ${\hat{g}}_{\mu \nu }\left(s\right)\equiv {\hat{g}}_{\mu \nu }\left(r\right)$. In addition, r=r(s) denotes the 4−position along a local field geodesic trajectory $C_{\left(r_{o},r_{1}\right)}$, which belongs to the background space-time $\left(Q^{4},\hat{g}\left(r\right)\right)$ and *s* is the Riemann length evaluated along a suitable family of field geodetics $\left\{C_{\left(r_{o},r_{1}\right)}\right\}$ prescribed so that for each $r\in Q^{4}$, by assumption there is a unique s∈I such that r=r(s). The prescription of *s* depends on the precise definition of the family of geodetics $\left\{C_{\left(r_{o},r_{1}\right)}\right\}$. As recalled above, this is established according to Ref. [23].

In Ref. [20], it was proved that ψ(s) is required to obey an evolution hyperbolic PDE, referred to as CQG-quantum-wave equation, determined in terms of suitable quantum Hamiltonian operators, namely of the form
(5)$$\frac{d\psi (s)}{ds}=\left[H^{\left(q\right)}\left(s\right),\psi \left(s\right)\right]=H^{\left(q\right)}\left(s\right)\psi \left(s\right).$$

Such an equation determines the proper-time evolution of ψ(s) along a field geodetics subject to suitable initial conditions of the type $\psi \left(s_{o}\right)=\psi_{o}\left(g,\hat{g}\left(s_{o}\right),s_{o},r\left(s_{o}\right)\right)$, with $\hat{g}\left(s_{o}\right)\equiv \hat{g}\left(r\left(s_{o}\right)\right)$ denoting the background metric tensor evaluated at initial position $r(s_{o})$. As shown elsewhere [23], it is possible to show that the CQG-quantum-wave Equation (5) can be equivalently cast in terms of a suitable set of quantum hydrodynamic equations upon introducing the Madelung representation for the wave function ψ(s). In the framework of CQG-theory one can show that these equations are realized by the so-called quantum continuity equation (see Appendix B) and a set of quantum Hamilton equations, which correspond to an equivalent quantum Hamilton–Jacobi equation (see again Ref. [23]).

Here we start our analysis by considering the initial-value problem associated with the quantum Hamilton equations, which are represented by the set of ODEs
(6)$$\left\{\begin{array}{c} \frac{dg_{\mu \nu }}{ds}=\frac{\partial H}{\partial \pi^{\mu \nu }},\\ \frac{d\pi^{\mu \nu }}{ds}=-\frac{\partial H}{\partial g_{\mu \nu }}, \end{array}\right.$$
with corresponding initial conditions
(7)$$\left\{\begin{array}{c} g_{\mu \nu }\left(s_{o}\right)=g_{\mu \nu }^{\left(o\right)}\left(r\left(s_{o}\right),s_{o}\right),\\ \pi^{\mu \nu }\left(s_{o}\right)=\pi^{\mu \nu (o)}\left(r\left(s_{o}\right),s_{o}\right). \end{array}\right.$$

Here the notation is standard. Thus, $x\equiv \left\{g_{\mu \nu }\left(s\right),\pi^{\mu \nu }\left(s\right)\right\}$ denotes the canonical quantum-hydrodynamic state, with
(8)$$\left\{\begin{array}{c} g_{\mu \nu }\left(s\right)=g_{\mu \nu }\left(r\left(s\right),s\right),\\ \pi^{\mu \nu }\left(s\right)=\pi^{\mu \nu }\left(r\left(s\right),s\right), \end{array}\right.$$
being the corresponding continuous Lagrangian coordinate and conjugate canonical momentum. Furthermore, $x\equiv \left\{g_{\mu \nu }\left(s\right),\pi^{\mu \nu }\left(s\right)\right\}$ and $H=H(x,\hat{g},r,s)$ denote now corresponding quantum functions, i.e., respectively the quantum canonical state and a suitable quantum 4−scalar Hamiltonian density (see below), with $g_{\mu \nu }\left(s\right)$ and $\pi^{\mu \nu }\left(s\right)$ that now identify the quantum gravitational field, namely the variational Lagrangian coordinate, and conjugate canonical momentum, respectively, both represented by second order 4−tensors. Equations (Equation 6) are defined with respect to the background space-time $\left\{Q^{4},\hat{g}\left(r\right)\right\}$, with $\hat{g}\left(r\right)\equiv \left\{{\hat{g}}_{\mu \nu }\left(r\right)\right\}\equiv \left\{{\hat{g}}^{\mu \nu }\left(r\right)\right\}$ denoting the background metric tensor, parametrized with respect to a suitable GR-frame $r\equiv \left\{r^{\mu }\right\}$, which raises and lowers tensor indices and prescribes the geometry of the same space-time. Furthermore, as indicated above, here r=r(s), being s∈I the proper time along a geodetics belonging to *r*. This means that in Equation (Equation 6) the partial derivatives with respect to $g_{\mu \nu }\equiv g_{\mu \nu }\left(s\right)$ and $\pi^{\mu \nu }\equiv \pi^{\mu \nu }\left(s\right)$ are performed keeping constant $\hat{g}\left(r\left(s\right)\right)$ (hereon denoted for brevity $\hat{g}\left(s\right)$) and all tensor functions of $\hat{g}\left(s\right)$ such as the Ricci tensor ${\hat{R}}^{\mu \nu }\equiv R^{\mu \nu }\left(\hat{g}\left(s\right)\right)$. Moreover, $\frac{d}{ds}$ denotes the covariant s−derivative whose definition, recalled in Appendix A (see Equation (Equation 120)), is such that by construction the equation
(9)$$\frac{d}{ds}\hat{g}\left(s\right)=0$$
holds identically. Finally, following Ref. [20], the quantum Hamiltonian density $H=H(x,\hat{g},r,s)$ is defined as
(10)$$H=T+V+V_{QM},$$
with *T* denoting the effective kinetic energy
(11)$$T=\frac{\pi^{\mu \nu }\pi_{\mu \nu }}{2\alpha L},$$
while $V=V\left(g,\hat{g},r,s\right)$ is the classical potential energy, namely
(12)$$\left\{\begin{array}{c} V\left(g,\hat{g},r,s\right)\equiv \sigma V_{o}\left(g,\hat{g}\right)+\sigma V_{F}\left(g,\hat{g},r,s\right),\\ V_{o}\left(g,\hat{g}\right)\equiv h\left(g\left(s\right)\right)\alpha Lg_{\mu \nu }\left(s\right){\hat{R}}^{\mu \nu },\\ V_{F}\left(g,\hat{g},r\right)\equiv hL_{F}\left(g,\hat{g},r\right), \end{array}\right.$$
with $V_{o}\left(g,\hat{g}\right)$ and $V_{F}\left(g,\hat{g},r\right)$ representing the classical vacuum and external effective potentials and σ=±1 denoting a signature factor to be properly determined. In the previous equations *h* denotes the variational weight factor
(13)$$h\left(g\left(s\right)\right)=2-\frac{1}{4}g_{\alpha \beta }\left(s\right)g^{\alpha \beta }\left(s\right),$$
which is a characteristic term of the synchronous variational principle at the basis of the manifestly covariant classical theory of GR and of CQG-theory (see Ref. [20]). Finally, $V_{QM}=V_{QM}\left(g,\hat{g},r,s\right)$ is the Bohm quantum effective potential
(14)$$V_{QM}\left(g,\hat{g},r,s\right)\equiv \sigma \frac{\hbar^{2}}{8\alpha L}\frac{\partial \ln\rho }{\partial g^{\mu \nu }}\frac{\partial \ln\rho }{\partial g_{\mu \nu }}-\sigma \frac{\hbar^{2}}{4\alpha L}\frac{\partial^{2}\rho }{\rho \partial g_{\mu \nu }\partial g^{\mu \nu }},$$
which arises when a Bohmian-like representation is adopted for the quantum-gravity wave function ψ in terms of the Madelung variables, so that the quantum-wave equation can be equivalently expressed by the couple of quantum-hydrodynamic equations formed by the continuity and momentum equations. As a result, the function $\rho \equiv \rho (\Delta g-\hat{g}\left(s\right))$ is the Gaussian quantum probability density function (PDF) that is a solution of the quantum continuity equation (see Appendix B) and is given by
(15)$$\rho \left(\Delta g-\hat{g}\left(s\right)\right)=\frac{1}{\langle \langle 1\rangle \rangle }\hat{\rho }\left(s\right)\exp\left\{-\frac{{\left(\Delta g\left(s\right)-\hat{g}\left(s\right)\right)}^{2}}{r_{th}^{2}}\right\},$$
with $\langle \langle 1\rangle \rangle$ and $\hat{\rho }\left(s\right)$ being a suitably prescribed (see Appendix B). Explicitly, the exponent ${\left(\Delta g\left(s\right)-\hat{g}\left(s\right)\right)}^{2}$ can be equivalently expressed as
(16)$${\left(\Delta g-\hat{g}\left(s\right)\right)}^{2}\equiv \left[\Delta g^{\alpha \beta }-{\hat{g}}^{\alpha \beta }\left(s\right)\right]\left[\Delta g_{{}^{\alpha \beta }}-{\hat{g}}_{{}^{\alpha \beta }}\left(s\right)\right].$$

Equation (Equation 14) can then be evaluated explicitly. This implies that the Bohm potential can be represented as $V_{QM}\left(g,\hat{g},r,s\right)\equiv V_{QM}\left(\Delta g-\hat{g}\left(s\right),\hat{g}\right)$ and is given by
(17)$$V_{QM}\left(\Delta g-\hat{g}\left(s\right),\hat{g}\right)=\frac{\sigma \hbar^{2}}{4\alpha L}\frac{8p^{2}\left(s\right)}{r_{th}^{2}}-\frac{\sigma }{2}\Lambda_{QM}^{\left(eff\right)}\left(s\right){\left(\Delta g-\hat{g}\left(s\right)\right)}^{2},$$
with
(18)$$\begin{array}{ccc} \Lambda_{QM}^{\left(eff\right)}\left(s\right) & \equiv & \Lambda_{QM}p^{3}\left(s\right), \end{array}$$
(19)$$\begin{array}{ccc} \Lambda_{QM} & \equiv & \frac{\hbar^{2}}{\alpha Lr_{th}^{4}}, \end{array}$$
denoting the effective and constant quantum cosmological constants, respectively, and with p(s) being a suitable quantum function, previously reported in Ref. [22] and recalled in Equation (Equation 126) of Appendix B. The rest of the notations is standard. Thus, *ℏ* is the reduced Planck constant, $r_{th}$ is a suitable dimensionless constant 4−scalar while, following Ref. [20], α and *L* are the dimensional constant $\alpha =m_{o}cL$ and *L* is the Compton length associated with the graviton mass $m_{o}$, respectively, namely $L=\frac{\hbar }{m_{o}c}$. As a consequence, the Hamilton Equations (Equation 6) written explicitly yield
(20)$$\left\{\begin{array}{c} \frac{dg_{\mu \nu }}{ds}=\frac{1}{\alpha L}\pi_{\mu \nu },\\ \frac{d\pi^{\mu \nu }}{ds}=-\frac{\partial \left(V(g\left(s\right),\hat{g}\left(s\right),r,s+V_{QM}\left(\Delta g-\hat{g}\left(s\right),\hat{g}\left(s\right)\right)\right)}{\partial g_{\mu \nu }}, \end{array}\right.$$
and thus they are equivalent to the Lagrange equations
(21)$$\frac{d^{2}g_{\mu \nu }}{ds^{2}}=-\frac{1}{\alpha L}\frac{\partial V\left(g,\hat{g},r,s\right)}{\partial g^{\mu \nu }}+B_{\mu \nu }\left(r,s\right).$$

Here, explicit evaluation of the partial derivative with respect to $g^{\mu \nu }\left(s\right)$ (and performed at constant $\hat{g}\left(s\right))$ delivers
(22)$$\left.-\frac{1}{\alpha L}\frac{\partial V\left(g,\hat{g},r,s\right)}{\partial g^{\mu \nu }}=-\sigma h\left(g\left(s\right)\right){\hat{R}}_{\mu \nu }+\frac{\sigma }{2}g_{\mu \nu }g^{ik}{\hat{R}}_{ik},\right.$$
while the second term on the rhs of the Lagrange equation, namely
(23)$$B_{\mu \nu }\left(r,s\right)\equiv -\frac{1}{\alpha L}\frac{\partial }{\partial g^{\mu \nu }}V_{QM}\left(g,\hat{g},r,s\right),$$
yields what is referred to as a Bohm source tensor field [23], namely
(24)$$B_{\mu \nu }\left(r,s\right)\equiv \frac{\sigma \Lambda_{QM}^{\left(eff\right)}\left(s\right)}{\alpha L}\left[\Delta g_{\mu \nu }-{\hat{g}}_{\mu \nu }\right].$$

### Background Equilibrium Solution of EFE

We remark that in the previous equations the background space-time metric tensor $\hat{g}\left(s\right)$ is not arbitrary. The equation that determines it follows, in fact, in a consistent manner from the same canonical equations stated above, i.e., Equation (Equation 3). As discussed in Ref. [23] it is obtained subject to the following requirements:

(A) The stationarity condition
(25)$$\frac{d}{ds}\hat{g}\left(s\right)=0,$$
i.e., the requirement that the conjugate momentum vanishes identically
(26)$$\hat{\pi }\left(s\right)\equiv \left\{{\hat{\pi }}_{\mu \nu }\left(s\right)\right\}\equiv \left\{{\hat{\pi }}^{\mu \nu }\left(s\right)\right\}\equiv 0.$$

(B) The extremum condition
(27)$${\left.\frac{\partial \left(V(g\left(s\right),\hat{g}\left(s\right),r,s+V_{QM}\left(\Delta g-\hat{g}\left(s\right),\hat{g}\left(s\right)\right)\right)}{\partial g^{\mu \nu }}\right|}_{g\left(s\right)=\hat{g}\left(s\right)}=0.$$

(C) Setting in Equation (Equation 27) also the extremal deterministic condition Δg≡0, namely
(28)$${\left.\frac{\partial \left(V(g\left(s\right),\hat{g}\left(s\right),r,s+V_{QM}\left(\Delta g-\hat{g}\left(s\right),\hat{g}\left(s\right)\right)\right)}{\partial g^{\mu \nu }}\right|}_{\begin{array}{c} g\left(s\right)=\hat{g}\left(s\right)\\ \Delta g\equiv 0 \end{array}}=0.$$

We notice in fact that Δg denotes the quantum stochastic displacement field associated with the quantum stochastic trajectories associated with the quantum PDF and driven by the Bohm potential, which characterizes the quantum field g(s). Thus for condition (A) to apply the Christoffel connections contained in the covariant derivative must be prescribed in terms of $\hat{g}\left(s\right)$. Assuming without loss of generality $V_{F}\left(g,\hat{g},r\right)=0$, namely the vacuum condition, from Equation (Equation 27) and condition (C) one obtains the second order PDE that identifies the quantum-modified EFE carrying the contribution of the quantum cosmological constant, namely
(29)$$-\sigma {\hat{R}}_{\mu \nu }+\frac{\sigma }{2}{\hat{g}}_{\mu \nu }\left(s\right)\hat{R}-\sigma {\hat{g}}_{\mu \nu }\left(s\right)\Lambda_{QM}^{\left(eff\right)}\left(s\right)=0,$$
where ${\hat{R}}_{\mu \nu }$$\equiv R_{\mu \nu }\left(\hat{g}\left(s\right)\right)$ and $\hat{R}\equiv R\left(\hat{g}\left(s\right)\right)$ denote the Ricci tensor and Ricci 4−scalar, respectively, both expressed in terms of $\hat{g}\left(s\right)$. Therefore this delivers
(30)$$\hat{R}\equiv R\left(\hat{g}\left(s\right)\right)=4\Lambda_{QM}^{\left(eff\right)}\left(s\right).$$

In particular, one can show (see Appendix B and Ref. [22]) that under suitable assumptions the function p(s) appearing in Equation (Equation 18) can be set equal to p(s)=1. In the following, for simplicity we shall ignore possible quantum effects of this type, thus setting p(s)=1. Hence, $\Lambda_{QM}^{\left(eff\right)}\left(s\right)$ reduces to the constant $\Lambda_{QM}^{\left(eff\right)}\left(s\right)=\Lambda_{QM}$, i.e., the constant quantum-produced CC which, in the absence of other classical effects (for example so-called gravitational sigma-models [30]) is predicted by CQG-theory [23]. Furthermore, we shall assume that the quantum cosmological constant is independent of *s* so that Equation (Equation 29) reduces to
(31)$$-\sigma {\hat{R}}_{\mu \nu }+\frac{\sigma }{2}{\hat{g}}_{\mu \nu }\left(s\right)\hat{R}-\sigma {\hat{g}}_{\mu \nu }\left(s\right)\Lambda_{QM}=0,$$
which implies in turn
(32)$$\hat{R}\equiv R\left(\hat{g}\left(s\right)\right)=4\Lambda_{QM}.$$

We stress that Equation (Equation 32) recovers exactly the Einstein field equation in vacuum for the background metric tensor field $\hat{g}\left(s\right)$. As a consequence CQG-theory embodies consistently all the relevant physics associated with EFE, such as the occurrence of BH’s and associated event horizons, as well as multiple scale effects when both CC and Newtonian scales are present, the latter being represented through the gravitational radius $GM/c^{2}$ [31].

## 3. Search of Non-Stationary Scale-Transformed Solutions

In this section, we set the mathematical framework for the construction of non-stationary solutions of the quantum Hamilton equations (i.e., see Equation (Equation 6)), in order to subsequently investigate whether they can provide a valuable route for the regularization of classical singularities in BH solutions (see also subsequent Section 6 and Section 7). As we intend to show, such solutions, unlike Equation (Equation 25) invoked above for the determination of the background MFT $\hat{g}\left(s\right)$, are characterized by a suitably-prescribed, non-vanishing and non-constant canonical momentum $\pi \equiv \left\{\pi_{\mu \nu }\right\}\equiv \left\{\pi^{\mu \nu }\right\}$. More precisely, the generalized coordinate are now sought of the (4−tensor) form
(33)$$\begin{array}{ccc} g^{\left(d\right)}\left(s\right) & = & N(s)g(r(s),s), \end{array}$$
(34)$$\begin{array}{ccc} {\hat{g}}^{\left(d\right)}\left(s\right) & = & N\left(s\right)\hat{g}\left(s\right), \end{array}$$
with N(s) denoting a suitable non-vanishing 4−scalar function of the proper time *s* and $\hat{g}\left(s\right)$ being the background metric tensor (Equation 3). We stress that here: (a) The same multiplicative factor N(s) occurs both in the covariant and in the counter-variant components, namely
(35)$$\left\{\begin{array}{c} g_{\mu \nu }^{\left(d\right)}\left(s\right)=N\left(s\right)g_{\mu \nu }\left(r\left(s\right),s\right),\\ g^{(d)\mu \nu }\left(s\right)=N\left(s\right)g^{\mu \nu }\left(r\left(s\right),s\right), \end{array}\right.$$
and
(36)$$\left\{\begin{array}{c} {\hat{g}}_{\mu \nu }^{\left(d\right)}\left(s\right)=N\left(s\right){\hat{g}}_{\mu \nu }\left(s\right),\\ {\hat{g}}^{(d)\mu \nu }\left(s\right)=N\left(s\right){\hat{g}}^{\mu \nu }\left(s\right); \end{array}\right.$$
(b) in the same Equations (Equation 35) and (Equation 36) all tensor indexes are raised and lowered by the background metric tensor $\hat{g}\left(s\right)$ only. Equations (Equation 33) and (34) represent respectively transformations of the quantum fluid field g(s)≡g(r(s),s) and the background field $\hat{g}\left(s\right)$ that are generated via the *scale transformation*
(37)$$\left\{\begin{array}{c} g\left(r\left(s\right),s\right)\to g^{\left(d\right)}\left(s\right)=N\left(s\right)g\left(r\left(s\right),s\right),\\ \hat{g}\left(s\right)\to {\hat{g}}^{\left(d\right)}\left(s\right)=N\left(s\right)\hat{g}\left(s\right). \end{array}\right.$$

For this reason the tensor field ${\hat{g}}^{\left(d\right)}\left(r\left(s\right)\right)$ and the scalar function N(s) (with N(s) still to be suitably determined) are referred to here as scale-transformed fields and scale form factor, respectively.

Let us now pose the problem of (determining) the proper-time evolution of the scale for factor N(s). The required prescription follows from the quantum Hamilton Equation (Equation 6) by introducing the scale transformation (Equation 37). As we intend to show now, this allows us to determine uniquely a constraint equation for the scalar form factor. Indeed the equation for ${\hat{g}}^{\left(d\right)}\left(s\right)$ follows from Equation (Equation 6) by introducing the replacements
(38)$$\left\{\begin{array}{c} g\left(r\left(s\right),s\right)\to N\left(s\right)g\left(r\left(s\right),s\right)\equiv g^{\left(d\right)}\left(s\right),\\ \Delta g(s)\to N(s)\Delta g(s),\\ \hat{g}\left(s\right)\to N\left(s\right)\hat{g}\left(s\right)\equiv {\hat{g}}^{\left(d\right)}\left(s\right),\\ h(g\left(r\left(s\right),s,\hat{g}\left(s\right)\right)\to h\left(N\left(s\right)g\left(r\left(s\right),s\right),\hat{g}\left(s\right)\right). \end{array}\right.$$

In particular, one needs to evaluate the corresponding ODEs holding for ${\hat{g}}_{\mu \nu }^{\left(d\right)}\left(s\right)$, which are implied by Equations (Equation 6) and (Equation 31) (and equivalently Equation (Equation 21)).The procedure to obtain it is analogous to that for $\hat{g}\left(s\right)$ (see related discussion in Ref. [23]). For later use, first one notices that multiplying Equation (Equation 31) term by term by N(s) one obtains
(39)$$-\sigma N\left(s\right){\hat{R}}_{\mu \nu }+\frac{\sigma }{2}{\hat{g}}_{\mu \nu }^{\left(d\right)}\left(s\right)\hat{R}-\sigma {\hat{g}}_{\mu \nu }^{\left(d\right)}\left(s\right)\Lambda_{QM}=0,$$
which implies the (obvious) conclusion that EFE determines uniquely also the scale-transformed field ${\hat{g}}_{\mu \nu }^{\left(d\right)}\left(s\right)$. Second, let us represent Equation (Equation 6) in terms of Equation (Equation 38). Upon first setting Δg=0, i.e., requiring the vanishing of the stochastic displacement tensor that characterizes the Bohm interaction term one obtains
(40)$$\left\{\begin{array}{c} \frac{dN\left(s\right)g_{\mu \nu }}{ds}=\frac{1}{\alpha L}\pi_{\mu \nu },\\ \frac{1}{\alpha L}\frac{d\pi_{\mu \nu }}{ds}=-\frac{1}{\alpha L}\frac{\partial V\left({\hat{g}}^{\left(d\right)}\left(s\right),\hat{g}\left(s\right),r,s\right)}{\partial g^{\mu \nu }}-\frac{\sigma \Lambda_{QM}}{\alpha L}N\left(s\right){\hat{g}}_{\mu \nu }. \end{array}\right.$$

Then, upon setting $g\left(s\right)=\hat{g}\left(s\right)$, we notice that unlike Equation (Equation 20), the canonical momentum $\pi_{\mu \nu }$ remains now non-vanishing and precisely such that
(41)$$\pi_{\mu \nu }\left(s\right)=\alpha L{\hat{g}}_{\mu \nu }\frac{dN(s)}{ds}.$$

Thus, the previous canonical equations now reduce to the Lagrange equations
(42)$${\hat{g}}_{\mu \nu }\frac{d^{2}N\left(s\right)}{ds^{2}}=-\frac{1}{\alpha L}\left.\frac{\partial V\left({\hat{g}}^{\left(d\right)}\left(s\right),\hat{g}\left(s\right),r,s\right)}{\partial g^{\mu \nu }}\right|-\frac{\sigma \Lambda_{QM}}{\alpha L}N\left(s\right){\hat{g}}_{\mu \nu }.$$

Direct evaluation of the rhs yields
(43)$$\begin{array}{ccc} & & \left.-\frac{1}{\alpha L}{\left.\frac{\partial V\left({\hat{g}}^{\left(d\right)}\left(s\right),\hat{g}\left(s\right),r,s\right)}{\partial g^{\mu \nu }}\right|}_{\begin{array}{c} g\left(s\right)=\hat{g}\left(s\right)\\ \Delta g=0 \end{array}}=-\sigma h\left(N\left(s\right)\hat{g}\left(s\right)\right)N\left(s\right){\hat{R}}_{\mu \nu }\right.\\ & & +\frac{\sigma }{2}{\hat{g}}_{\mu \nu }N^{3}\left(s\right)g^{ik}{\hat{R}}_{ik}, \end{array}$$
where σ=±1 is a still undetermined signature factor to be determined below (see next sections), while the variational factor becomes
(44)$$h\left(N\left(s\right)\hat{g}\left(s\right)\right)=2-\frac{1}{4}N^{2}\left(s\right){\hat{g}}^{\alpha \beta }\left(s\right){\hat{g}}_{\alpha \beta }\left(s\right)=2-N^{2}\left(s\right).$$

Equation (Equation 42) thus delivers
(45)$${\hat{g}}_{\mu \nu }\left(s\right)\frac{d^{2}N\left(s\right)}{ds^{2}}=\sigma \left[N^{2}\left(s\right)-2\right]N\left(s\right){\hat{R}}_{\mu \nu }+\frac{\sigma }{2}N^{3}\left(s\right){\hat{g}}_{\mu \nu }\left(s\right)\hat{R}-\sigma N\left(s\right)\Lambda_{QM}{\hat{g}}_{\mu \nu }\left(s\right),$$
where ${\hat{g}}_{\mu \nu }\left(s\right)$ satisfies by construction the Einstein field Equation (Equation 39). We intend to show that such an equation is integrable by quadratures.

## 4. Proper-Time Evolution Equation of the Scale-Form Factor N(s)

In this section we explicitly determine the proper-time evolution of the scale-form factor N(s) indicated above. For this purpose, to illustrate the procedure we assume first the case of Kottler (i.e., Schwarzschild–deSitter) metric space-time. This in fact is a representative solution that can be extended later to other BH configurations. In such a setting one notices that by construction $\hat{R}=4\Lambda_{QM}$.Then, saturation by ${\hat{g}}_{\mu \nu }$ yields from Equation (Equation 45)
(46)$$\frac{d^{2}N\left(s\right)}{ds^{2}}=-3\sigma \left[1-N^{2}\left(s\right)\right]N\left(s\right)\Lambda_{QM}.$$

Such an equation, subject to the prescription of the initial conditions $\left\{N\left(s_{o}\right),{\left.\frac{dN(s)}{ds}\right|}_{s_{o}}\right\}$ determines the proper-time evolution of the scale form-factor N(s).

Let us briefly point out its crucial qualitative properties.

First, we notice that $\frac{d^{2}N\left(s\right)}{ds^{2}}\to 0$ either if $N^{2}\left(s\right)\to 1$ or N(s)→0. The case N(s)=1 corresponds the standard background solution $\hat{g}\left(r\right)$ of EFE (see above Equation (Equation 29)). Notice that although by assumption N(s)≠0, nevertheless it can become infinitesimal (so that N(s)→0). This property, as shall be clarified below, will become crucial for the regularization of singular BH solutions.Second, the same Equation (Equation 46) is conservative. As a consequence it can therefore be reduced by a quadrature to an equivalent first order ODE. In fact it delivers:
(47)$$\frac{dN(s)}{ds}\frac{d^{2}N\left(s\right)}{ds^{2}}=\frac{d}{ds}\frac{1}{2}{\left[\frac{dN(s)}{ds}\right]}^{2}=3\sigma \frac{dN(s)}{ds}\left[N^{2}\left(s\right)-1\right]N\left(s\right)\Lambda_{QM}=\frac{3\sigma }{4}\frac{d}{ds}\left[N^{2}\left(s\right)\left(N^{2}\left(s\right)-2\right)\Lambda_{QM}\right],$$
which yields
(48)$$\frac{1}{2}{\left[\frac{dN(s)}{ds}\right]}^{2}-\frac{1}{2}{\left[\frac{dN(s)}{ds}\right]}_{s=s_{o}}^{2}=\frac{3\sigma }{4}\left[N^{2}\left(s\right)\left(N^{2}\left(s\right)-2\right)\Lambda_{QM}\right]-\frac{3\sigma }{4}\left[N^{2}\left(s_{o}\right)\left(N^{2}\left(s_{o}\right)-2\right)\Lambda_{QM}\right].$$As a consequence, setting the initial constant
(49)$$E={\left[\frac{dN(s)}{ds}\right]}_{s=s_{o}}^{2}-\frac{3\sigma }{2}\left[N^{2}\left(s_{o}\right)\left(N^{2}\left(s_{o}\right)-2\right)\Lambda_{QM}\right],$$Equation (Equation 46) yields the two possible ODE solutions
(50)$$\frac{dN(s)}{ds}=\pm \sqrt{E+\frac{3\sigma }{2}\left[N^{2}\left(s\right)\left(N^{2}\left(s\right)-2\right)\Lambda_{QM}\right]},$$
which are again solvable by quadratures. Notice that the requirement E=0 for $N(s_{o})=1$ requires setting also
(51)$${\left[\frac{dN(s)}{ds}\right]}_{s=s_{o}}=\pm \sqrt{-\frac{3\sigma }{2}\Lambda_{QM}},$$
where for the reality of ${\left[\frac{dN(s)}{ds}\right]}_{s=s_{o}}$necessarily one must set σ=−1, while the signature of the square root depends on whether the solution for $s>s_{o}$ is considered as growing or decaying. Notice, in particular, that if $N(s_{o})=1$ on a given EH, and if the orientation of the proper time *s* axis changes sign across the same EH, then the signature of the root $\pm \sqrt{\frac{3\sigma }{2}\left[\Lambda_{QM}\right]}$ should change sign across the same EH. As a consequence, an internally decaying (growing) solution should change to a growing (decaying) one outside. However, to determine the precise asymptotic behavior of $\frac{dN(s)}{ds}$ and N(s) a suitable classification must be adopted. The problem is analyzed separately in the following section.

## 5. Qualitative Properties of the Solutions

Let us now investigate in detail the qualitative properties of the solutions of Equation (Equation 46). Three cases are distinguished:Monotonically decaying solution in the inner BH domain.Monotonically growing/decaying solutions in the intermediate domain between two EH’s ($N^{2}\left(s_{o}\right)>1$).Monotonically decaying solution in the exterior BH domain.

A preliminary remark concerns the prescription of the signature factor σ=±1. The same factor appears in the prescription of the effective potential *V* (see Equation (Equation 12)) and (consequently) in the proper-time evolution equation for the scale-form factor N(s) (Equations (Equation 46) and (Equation 50)). In Ref. [20], the choice σ=−1 was adopted. Indeed this was shown to permit the existence of a discrete spectrum for the stationary CQG-wave equation and consequently the existence of a ground-state mass estimate for the graviton. The same prescription is introduced here for consistency. As we shall see, besides appropriate regularity conditions, it warrants the existence of monotonically decreasing/increasing solutions for N(s) to be pointed out below.

### 5.1. Monotonically Decaying Solution in the Inner BH Domain

Let us first consider the case of the interior domain of a BH in the case of a positive cosmological constant (CC). We seek a monotonically-decreasing solution for N(s) which vanishes asymptotically in r=0 and such that both $\frac{d^{2}N\left(s\right)}{ds^{2}}$ and $\frac{dN(s)}{ds}$ vanish in turn asymptotically only when also N(s)→0. Direct inspection shows that this actually requires initial conditions such that:

(a) E=0, namely such that ${\left[\frac{dN(s)}{ds}\right]}_{s=s_{o}}^{2}+\frac{3\sigma }{2}N^{2}\left(s_{o}\right)\left(N^{2}\left(s_{o}\right)-2\right)\Lambda_{QM}=0$;

(b) $N^{2}\left(s_{o}\right)<2$,
where in view of the prescription indicated above σ=−1. Incidentally we notice that such a choice is necessary for the validity of condition (a). Hence, in validity of the initial condition (b), $\frac{dN(s)}{ds}$ becomes
(52)$$\frac{dN(s)}{ds}=-\sqrt{\frac{3}{2}\left[N^{2}\left(s\right)\left(2-N^{2}\left(s\right)\right)\Lambda_{QM}\right]},$$
which implies that N(s) is indeed monotonically decreasing to zero for s→+∞, so that for all $s>s_{o}$, $N^{2}\left(s\right)<2$. The previous equation can then be integrated by quadratures yielding
(53)$$\int_{N(s_{o})}^{N(s)}\frac{dN}{N\sqrt{\frac{3}{2}\left[\left(2-N^{2}\right)\Lambda_{QM}\right]}}=s_{o}-s,$$
which in the limit for s→∞ yields
(54)$$\lim_{s\to +\infty }\int_{N(s_{o})}^{N(s)}\frac{dN}{N\sqrt{\frac{3}{2}\left(2-N^{2}\right)\Lambda_{QM}}}=-\infty .$$

Since N(s) decays to zero it follows that
(55)$$\lim_{s\to +\infty }\int_{N(s_{o})}^{N(s)}\frac{dN}{N\sqrt{\frac{3}{2}\left(2-N^{2}\right)\Lambda_{QM}}}∼\lim_{s\to +\infty }L_{QM}\ln N\left(s\right)=-\infty ,$$
with
(56)$$\begin{array}{ccc} & & \left.L_{QM}\equiv \frac{1}{\sqrt{3\Lambda_{QM}}}=\frac{1}{\sqrt{3\times 1.2\times 10^{-52}}}\;m≅0.527\times 10^{26}\;m,\right. \end{array}$$
(57)$$\begin{array}{ccc} & & \left.L_{\mathrm{deSitter}}=3L_{QM}\right., \end{array}$$
where $L_{QM}$ and $L_{\mathrm{deSitter}}$ denote respectively the CC characteristic scale length and the deSitter radius. This yields more precisely for s→∞
(58)$$L_{QM}\ln N\left(s\right)∼-s.$$

This proves therefore that the asymptotic behavior of the form factor N(s) for s→∞ is that of an exponential decay:(59)$$N\left(s\right)∼\exp\left(-s/L_{QM}\right),$$
which therefore occurs on the characteristic scale length $L_{QM}$.

Here we consider, as model test examples:

(1) The case of a two-parameter Kottler–Schwarzschild–deSitter space-time endowed with two EH’s (i.e., the inner Schwarzschild and the boundary deSitter EH’s, respectively) and a positive CC, with line element given by
(60)$$\begin{array}{ccc} ds^{2} & = & \alpha \left(r\right)c^{2}dt^{2}-\frac{1}{\alpha (r)}dr^{2}-r^{2}d\Omega^{2}, \end{array}$$
(61)$$\begin{array}{ccc} \alpha (r) & = & 1-\frac{r_{s}}{r}-\frac{\Lambda_{QM}r^{2}}{3}. \end{array}$$

(2) The case of a three-parameter Reissner–Nordstrom–deSitter space-time (charged BH and two EH’s), having line element
(62)$$\begin{array}{ccc} ds^{2} & = & \alpha \left(r\right)c^{2}dt^{2}-\frac{1}{\alpha (r)}dr^{2}-r^{2}d\Omega^{2}, \end{array}$$
(63)$$\begin{array}{ccc} \alpha (r) & = & 1-\frac{r_{s}}{r}+\frac{r_{Q}^{2}}{r^{2}}-\frac{\Lambda_{QM}r^{2}}{3}, \end{array}$$
(64)$$\begin{array}{ccc} r_{Q}^{2} & = & \frac{GQ^{2}}{4\pi \varepsilon_{0}c^{4}}, \end{array}$$
with *G* and $1/4\pi \varepsilon_{0}$ being the universal gravitational constant and the Coulomb interaction coupling constant, respectively.

(3) The case of a four-parameter FLRW–Schwarzschild–deSitter space-time (three EH’s), with line element
(65)$$\begin{array}{ccc} ds^{2} & = & \alpha \left(r\right)c^{2}dt^{2}-\frac{R(t)}{\beta (r)}\left[dr^{2}-\alpha \left(r\right)r^{2}d\Omega^{2}\right], \end{array}$$
(66)$$\begin{array}{ccc} \beta (r) & = & {\left(1+{\frac{kr}{4}}^{2}\right)}^{2}-\frac{r_{s}}{r}-\frac{\Lambda_{QM}r^{2}}{3}, \end{array}$$
(67)$$\begin{array}{ccc} \alpha (r) & = & 1-\frac{r_{s}}{r}-\frac{\Lambda_{QM}r^{2}}{3}. \end{array}$$

For the three cases indicated above, we consider in detail the internal problem, i.e., in the domain inside the inner EH. We intend to prove that for all such space-times the following asymptotic limit holds
(68)$$\lim_{s\to +\infty }N\left(s\right){\hat{g}}_{\mu \nu }\left(r\left(s\right)\right)\;∼\exp\frac{nL_{QM}}{r_{s}},$$
where $\frac{L_{QM}}{r_{s}}≅0.527\times 10^{23}/r_{s}$ (km), being $r_{s}$ measured in km and n=1 in cases (1) and (3), while n=2 in case (2). As a consequence in all such cases the conformally modified solution $N\left(s\right){\hat{g}}_{\mu \nu }$ is necessarily regular in the origin r=0.

The proof is as follows. The Riemann distance in the case of the quantum-modified Kottler solution $N\left(s\right){\hat{g}}_{\mu \nu }\left(r\left(s\right)\right)$ is obtained letting
(69)$$ds^{2}=N\left(s\right)\alpha \left(r\right)c^{2}dt^{2}-\frac{N(s)}{\alpha (r)}dr^{2}-N\left(s\right)r^{2}d\Omega^{2},$$
where one can set ${\left(\frac{rd\Omega }{ds}\right)}^{2}\equiv 0$ in the case of purely radial displacements. In the limit s→∞ for pure radial motion it therefore follows that
(70)$$1=N\left(s\right)\alpha \left(r\right){\left(\frac{cdt}{ds}\right)}^{2}-\frac{N(s)}{\alpha (r)}{\left(\frac{dr}{ds}\right)}^{2}∼\frac{r_{s}}{r(s)}{\left(\frac{cdt}{ds}\right)}^{2},$$
where assuming without loss of generality ${\left(cdt\right)}^{2}∼{\left(dr\right)}^{2}$ the contribution of the second term on the rhs is negligible, so that necessarily in the same limit $N\left(s\right)\alpha \left(r\right){\left(\frac{cdt}{ds}\right)}^{2}∼1$. This means that in a neighborhood of the origin r=0, *s* becomes infinite and as a consequence
(71)$$\left.\lim_{s\to \infty }\frac{r(s)}{r_{s}}=0.\right.$$

This means that asymptotically for $s\to \infty ,\frac{r(s)}{r_{s}}$ is an infinitesimal. Analogous conclusion holds in case (2), where instead it occurs
(72)$$\left.\lim_{s\to \infty }\frac{r^{2}\left(s\right)}{r_{Q}^{2}}=0.\right.$$

To estimate the asymptotic behavior of the product N(s)α(r) in the neighborhood of the r(s)=0 let us estimate asymptotically, for greater generality, the ratio
(73)$$\left|\frac{\exp(-s/L_{QM})}{{\left(\frac{r(s)}{R_{s}}\right)}^{n}}\right|∼\left|\frac{{\left(\frac{R_{s}}{r(s)}\right)}^{n}}{\exp(s/L_{QM})}\right|,$$
with n≥1 being an arbitrary real exponential and $R_{s}$ identifying respectively $r_{s}$ or $r_{Q}$. The asymptotic estimate (68) follows by taking the logarithm of numerator and denominator and upon differentiating them. It follows
(74)$$\left|\frac{n\ln\frac{R_{s}}{r(s)}}{\ln\exp(s/L_{QM})}\right|∼\left|\frac{n\frac{1}{r(s)}\frac{dr(s)}{ds}}{\frac{1}{L_{QM}}}\right|.$$

But since $\frac{r(s)}{R_{s}}∼\frac{dr(s)}{ds}$, it follows that
(75)$$\left|\frac{\ln{\left(\frac{R_{s}}{r(s)}\right)}^{n}}{\ln\exp(s/L_{QM})}\right|∼\frac{nL_{QM}}{R_{s}},$$
which implies Equation (Equation 68). The conclusion is therefore that in all cases indicated above for the background metric field tensor ${\hat{g}}_{\mu \nu }\left(r\right)$, the scale-transformed field $g_{\mu \nu }^{\left(d\right)}\left(s\right)=N\left(s\right){\hat{g}}_{\mu \nu }$ remains regular in the origin r=0 in the sense that in an arbitrary GR-frame and for all μ,ν=0,3:(76)$$\lim_{s\to \infty }\left|N\left(s\right){\hat{g}}_{\mu \nu }\left(r\left(s\right)\right)\right|∼\lim_{s\to \infty }∼\frac{nL_{QM}}{R_{s}}<\infty ,$$
where *n* is a suitable integer, $L_{QM}$ is the characteristic length (Equation 56) and $R_{s}$ is identified with the invariant characteristic length scales $r_{s}$ or $r_{Q}$, respectively. In particular, in the case of the conformally modified Kottler solution one obtains $\frac{nL_{QM}}{R_{s}}\equiv \frac{L_{QM}}{r_{s}}$ with n=1. Analogous conclusions hold for the Reissner–Nordstrom–deSitter space-time in which $\frac{nL_{QM}}{R_{s}}\equiv \frac{2L_{QM}}{r_{Q}}$ where again n=2, as well in the case of FLRW–Schwarzschild–deSitter space-time (provided the expansion coefficient R(t) remains strictly positive), where again $\frac{nL_{QM}}{R_{s}}\equiv \frac{L_{QM}}{r_{s}}$. Nevertheless, since in all cases considered here the factor $\frac{nL_{QM}}{R_{s}}$ is ≫1, the same conformal field is strongly peaked in the origin r=0.

### 5.2. Monotonically Growing/Decaying Solutions in the Intermediate Domain Between Two EH’s ($N^{2}\left(s_{o}\right)>1$)

Let us now consider the case of the intermediate region between two EH’s (intermediate domain problem). A prototype of such an occurrence is the Kottler space-time, characterized by a inner Schwarzschild EH and external deSitter EH. We intend to show that both growing and decaying monotonic solutions exist.

Let us consider first the case of a growing solution. For definiteness, let us require that $s_{o}$ and $s_{1}$ denote the initial and final proper times along a geodetics, with $r\left(s_{o}\right)=\rho_{o}$ and $r\left(s_{1}\right)=\rho_{1}$ denoting the initial and final radii (of the same curve) with $r_{o}$ and $r_{1}$, respectively, assumed to be suitably close to the two EH, namely such that
(77)$$\begin{array}{ccc} r_{1} & < & \rho_{o}<r_{1}+\varepsilon^{2}, \end{array}$$
(78)$$\begin{array}{ccc} L_{\mathrm{deSitter}}-\varepsilon^{2} & < & \rho_{o}<L_{\mathrm{deSitter}}+\varepsilon^{2}, \end{array}$$
such that they are located outside the radius $r_{1}$ of the Schwarzschild’s EH and inside the corresponding radius of the deSitter EH. Setting again σ=−1, let us then consider a first integral of the form
(79)$$\frac{dN(s)}{ds}=\sqrt{E-\frac{3}{2}N^{2}\left(s\right)\left(N^{2}\left(s\right)-2)\right)\Lambda_{QM}},$$
which is assumed subject to the initial conditions such that
(80)$$\left\{\begin{array}{c} N^{2}\left(s_{o}\right)=1+\Delta^{2},\\ {\left[\frac{dN(s)}{ds}\right]}_{s=s_{o}}^{2}=E-\frac{3\Lambda_{QM}}{2}\left(1+\Delta^{2}\right)\left(\Delta^{2}-1)\right)=E-\frac{3\Lambda_{QM}}{2}\left(\Delta^{4}-1\right)\geq 0, \end{array}\right.$$
and where $\Delta^{2}$ is in principle an arbitrary real number such that ${\left[\frac{dN(s)}{ds}\right]}_{s=s_{o}}^{2}\geq 0$. It follows that for $s>s_{o}$ the solution of Equation (Equation 79) N(s) is necessarily monotonically growing for $E-\frac{3}{2}N^{2}\left(s\right)\left(N^{2}\left(s\right)-2\right)\Lambda_{QM}\geq 0$ because then $\frac{dN(s)}{ds}\geq 0$, but also bounded and such that
(81)$$1\leq N\left(s_{o}\right)\leq N\left(s\right)\leq N_{\max}.$$

Here the upper bound $N_{\max}$ depends on the initial “energy” *E*, being such that
(82)$$N_{\max}=1+\sqrt{1+\frac{2E}{3\Lambda_{QM}}}.$$

As an example, setting Δ=0 in Equation (Equation 80) and $E=-3\Lambda_{QM}/2$ this implies that identically $\frac{dN(s)}{ds}=0$ and $N\left(s_{o}\right)=N\left(s\right)=N_{\max}=1$. We stress that provided the said initial “energy” *E* is suitably prescribed then $N_{\max}$ can become arbitrarily large. This happens provided
(83)$$\sqrt{1+\frac{2E}{3\Lambda_{QM}}}\gg 1\Rightarrow N_{\max}\gg 1.$$

As we shall see below this is equivalent to an inflationary condition. Thus, we conclude that in validity of the initial conditions (Equation 80), Equation (Equation 79) determines a monotonically increasing solution with energy-dependent upper bound $N_{\max}$.

Instead the other root
(84)$$\frac{dN(s)}{ds}=-\sqrt{E-\frac{3}{2}\left[N^{2}\left(s\right)\left(N^{2}\left(s\right)-2)\right)\Lambda_{QM}\right]}$$
corresponds to a decreasing solution, of the type
(85)$$N_{\min}\leq N\left(s\right)\leq N\left(s_{o}\right)\leq N_{\max},$$
where
(86)$$N_{\min}=1-\sqrt{1+\frac{2E}{3\Lambda_{QM}}}.$$
can become negative. Thus we conclude that in validity of the initial conditions (Equation 80), Equation (Equation 84) determines a mononically decreasing solution with energy-dependent lower bound. Such a solution becomes negative if $\sqrt{1+\frac{2E}{3\Lambda_{QM}}}>1$. However, if one demands that N(s) remains strictly positive such a solution should be considered nonphysical.

### 5.3. Monotonically Decaying Solution in the Exterior BH Domain

Let us now consider the exterior problem, i.e., in the domain outside the EH of a BH, assuming that the external domain is infinite (i.e., that no other EH is present). We claim that, setting again σ=−1, in such a case admissible solutions are again bounded as in the intermediate case considered above. As a consequence, ruling out nonphysical decaying solutions in which N(s) vanishes or becomes negative, it follows that for s→∞, N(s) is of the type given by Equation (Equation 79). N(s) is therefore necessarily monotonically growing for $E-\frac{3}{2}N^{2}\left(s\right)\left(N^{2}\left(s\right)-2)\right)\Lambda_{QM}\geq 0$ and is bounded because $N\left(s\right)=N_{\max}$ for s→∞.

## 6. Construction of Background Conformal MFT Solutions

In this section we intend to prove that the (covariant and countervariant) representations ${\hat{g}}_{\mu \nu }^{\left(d\right)}\left(s\right)$ and ${\hat{g}}^{(d)\mu \nu }\left(s\right)$ given above (see Equation (Equation 36)) for the scale-transformed field ${\hat{g}}^{\left(d\right)}\left(s\right)$ actually allow the realization of two different space-time conformal tensor fields, here labeled as ${\hat{g}}^{\left(C\right)}\left(s\right)$ and ${\hat{g}}^{\left(C_{1}\right)}\left(s\right)$, respectively. The question is whether the same tensor fields can also be viewed as representing admissible realizations of the background space-time metric field tensor (MFT).

First, let us consider the prescription of the fields ${\hat{g}}^{\left(C\right)}\left(s\right)$ and ${\hat{g}}^{\left(C_{1}\right)}\left(s\right)$: this based on Equation (Equation 36), whereby the tensor fields ${\hat{g}}_{\mu \nu }^{\left(C\right)}\left(s\right)$ and ${\hat{g}}_{\mu \nu }^{\left(C_{1}\right)}\left(s\right)$ are prescribed as follows
(87)$$\left\{\begin{array}{c} {\hat{g}}_{\mu \nu }^{\left(C\right)}\left(s\right)=N\left(s\right){\hat{g}}_{\mu \nu }\left(s\right),\\ {\hat{g}}^{(C)\mu \nu }\left(s\right)=\frac{1}{N(s)}{\hat{g}}^{\mu \nu }\left(s\right), \end{array}\right.$$
and respectively
(88)$$\left\{\begin{array}{c} {\hat{g}}_{\mu \nu }^{\left(C_{1}\right)}\left(s\right)=\frac{1}{N(s)}{\hat{g}}_{\mu \nu }\left(s\right),\\ {\hat{g}}^{(C_{1})\mu \nu }\left(s\right)=N\left(s\right){\hat{g}}^{\mu \nu }\left(s\right). \end{array}\right.$$

As an obvious consequence it then follows that both ${\hat{g}}^{\left(C\right)}\left(s\right)$ and ${\hat{g}}^{\left(C_{1}\right)}\left(s\right)$ are conformal fields [32] satisfying the orthogonality conditions
(89)$$\left\{\begin{array}{c} {\hat{g}}_{\mu \alpha }^{\left(C\right)}\left(s\right){\hat{g}}^{(C)\mu \beta }\left(s\right)=\delta_{\alpha }^{\beta },\\ {\hat{g}}_{\mu \alpha }^{\left(C_{1}\right)}\left(s\right){\hat{g}}^{(C_{1})\mu \beta }\left(s\right)=\delta_{\alpha }^{\beta }. \end{array}\right.$$

We stress that analogous conformal representation can be obtained also for the generic quantum field g(s) as well for the stochastic quantum displacement field Δg, in terms of their covariant and counter-variant components, namely $g_{\mu \nu }\left(s\right),\Delta g_{\mu \nu }$ and $g^{\mu \nu }\left(s\right),\Delta g^{\mu \nu }$, respectively, thus yielding in particular
(90)$$\left\{\begin{array}{c} g_{\mu \nu }^{\left(C\right)}\left(s\right)=N\left(s\right)g_{\mu \nu }\left(r\left(s\right),s\right),\\ g^{(C)\mu \nu }\left(s\right)=\frac{1}{N(s)}g^{\mu \nu }\left(r\left(s\right),s\right), \end{array}\right.$$
and
(91)$$\left\{\begin{array}{c} \Delta g_{\mu \nu }^{\left(C\right)}\left(s\right)=N\left(s\right)\Delta g_{\mu \nu },\\ \Delta g^{(C)\mu \nu }\left(s\right)=\frac{1}{N(s)}\Delta g^{\mu \nu }. \end{array}\right.$$

### 6.1. Conformal Riemann Tensor, Ricci Tensor and 4−Scalar

As a further step, let us proceed identifying in each case the corresponding (i.e., here referred to as “*conformal*”) Riemann and Ricci 4−tensors as well as the corresponding Ricci scalars, i.e., $R_{\sigma \mu \nu }^{\rho }\left({\hat{g}}^{\left(C\right)}\left(s\right)\right)$, $R_{\sigma \mu \nu }^{\rho }\left({\hat{g}}^{\left(C_{1}\right)}\left(s\right)\right)$, $R_{\mu \nu }\left({\hat{g}}^{\left(C\right)}\left(s\right)\right)$, $R_{\rho \mu \nu }^{\rho }\left({\hat{g}}^{\left(C\right)}\left(s\right)\right)$ and finally $R({\hat{g}}^{\left(C\right)}\left(s\right))$ and $R({\hat{g}}^{\left(C_{1}\right)}\left(s\right))$, respectively. Here, we wish to determine the relationships holding among them. Such relationships are in fact relevant to assess their physical interpretation and in particular for the identification of the corresponding Einstein field equations holding for them, in analogy with Equation (Equation 31), which applies for the background metric tensor field $\hat{g}\left(s\right)$. The determination of such relationship is actually straightforward. Let us start noting, in fact, that by construction, the Christoffel symbols satisfy the invariance property:(92)$$\Gamma_{\nu \sigma }^{\rho }\left(\hat{g}\left(s\right)\right)=\Gamma_{\nu \sigma }^{\rho }\left({\hat{g}}^{\left(C\right)}\left(s\right)\right)=\Gamma_{\nu \sigma }^{\rho }\left({\hat{g}}^{\left(C_{1}\right)}\left(s\right)\right).$$

As a consequence also the Riemann tensor
(93)$$R_{\sigma \mu \nu }^{\rho }\left(\hat{g}\left(s\right)\right)=\partial_{\mu }\Gamma_{\nu \sigma }^{\rho }-\partial_{\nu }\Gamma_{\mu \sigma }^{\rho }+\Gamma_{\mu \lambda }^{\rho }\Gamma_{\nu \sigma }^{\lambda }-\Gamma_{\nu \lambda }^{\rho }\Gamma_{\mu \sigma }^{\lambda }$$
is similarly invariant, since by construction it then follows that
(94)$$\left\{\begin{array}{c} R_{\sigma \mu \nu }^{\rho }\left({\hat{g}}^{\left(C\right)}\left(s\right)\right)=R_{\sigma \mu \nu }^{\rho }\left(\hat{g}\left(s\right)\right),\\ R_{\sigma \mu \nu }^{\rho }\left({\hat{g}}^{\left(C_{1}\right)}\left(s\right)\right)=R_{\sigma \mu \nu }^{\rho }\left(\hat{g}\left(s\right)\right). \end{array}\right.$$

Therefore, the same invariance property occurs also for the covariant components of the Ricci tensor, namely
(95)$$\left\{\begin{array}{c} R_{\mu \nu }\left({\hat{g}}^{\left(C\right)}\left(s\right)\right)=R_{\rho \mu \nu }^{\rho }\left({\hat{g}}^{\left(C\right)}\left(s\right)\right),\\ R_{\mu \nu }\left({\hat{g}}^{\left(C_{1}\right)}\left(s\right)\right)=R_{\rho \mu \nu }^{\rho }\left({\hat{g}}^{\left(C_{1}\right)}\left(s\right)\right). \end{array}\right.$$

Therefore, this implies that the Ricci 4−scalars satisfy the relationships
(96)$$\left\{\begin{array}{c} R\left({\hat{g}}^{\left(C\right)}\left(s\right)\right)={\hat{g}}^{(C)\mu \nu }\left(s\right)R_{\mu \nu }\left({\hat{g}}^{\left(C\right)}\left(s\right)\right)=\frac{R(\hat{g}\left(s\right))}{N(s)},\\ R\left({\hat{g}}^{\left(C_{1}\right)}\left(s\right)\right)={\hat{g}}^{(C_{1})\mu \nu }\left(s\right)R_{\mu \nu }\left({\hat{g}}^{\left(C\right)}\left(s\right)\right)=N\left(s\right)R\left(\hat{g}\left(s\right)\right), \end{array}\right.$$
which imply in turn necessarily the prescription of suitably scaled-down (or increased) Ricci 4−scalars
(97)$$\left\{\begin{array}{c} R\left({\hat{g}}^{\left(C\right)}\left(s\right)\right)=\frac{1}{N(s)}R\left(\hat{g}\left(s\right)\right),\\ R\left({\hat{g}}^{\left(C_{1}\right)}\left(s\right)\right)=N\left(s\right)R\left(\hat{g}\left(s\right)\right). \end{array}\right.$$

Thus, denoting conventionally $\Lambda_{QM}\left(\hat{g}\right)$ the cosmological constant defined by Equation (19) and based on Equation (Equation 32) one obtains
(98)$$\left\{\begin{array}{c} R\left({\hat{g}}^{\left(C\right)}\left(s\right)\right)=\frac{4\Lambda_{QM}\left(\hat{g}\left(s\right)\right)}{N(s)}\equiv 4\Lambda_{QM}\left({\hat{g}}^{\left(C\right)}\left(s\right)\right),\\ R\left({\hat{g}}^{\left(C_{1}\right)}\left(s\right)\right)=4N\left(s\right)\Lambda_{QM}\left(\hat{g}\left(s\right)\right)\equiv 4\Lambda_{QM}\left({\hat{g}}^{\left(C_{1}\right)}\left(s\right)\right). \end{array}\right.$$

This implies in turn for consistency that also the cosmological constant must be, in the two cases, suitably scaled-down or increased according to the prescription
(99)$$\Lambda_{QM}\left({\hat{g}}^{\left(C\right)}\left(s\right)\right)=\frac{\Lambda_{QM}\left(\hat{g}\left(s\right)\right)}{N(s)},$$
(100)$$\Lambda_{QM}\left({\hat{g}}^{\left(C_{1}\right)}\left(s\right)\right)=N\left(s\right)\Lambda_{QM}\left(\hat{g}\left(s\right)\right).$$

This shows that:The effective quantum CC $\Lambda_{QM}\left({\hat{g}}^{\left(C\right)}\left(s\right)\right)$ is actually scaled-down by the factor $\frac{1}{N(s)}$. This means that the effective cosmological constant that characterizes the quantum-modified equilibrium ${\hat{g}}_{\mu \nu }^{\left(C\right)}\left(s\right)$ actually diverges when N(s)→0.Conversely, instead, the effective CC $\Lambda_{QM}\left({\hat{g}}^{\left(C_{1}\right)}\left(s\right)\right)$ is actually increased by the factor N(s). This means that the effective cosmological constant that characterizes the alternate (regular) quantum-modified equilibrium ${\hat{g}}^{(C_{1})\mu \nu }\left(s\right)=N\left(s\right){\hat{g}}^{\mu \nu }\left(r\left(s\right)\right)$ and its conformally conjugate metric tensor ${\hat{g}}_{\mu \nu }^{\left(C_{1}\right)}\left(s\right)=\frac{1}{N(s)}{\hat{g}}^{\mu \nu }\left(r\left(s\right)\right)$ actually tends to zero when N(s)→0.

However, the most relevant physical aspect concerns the regularity of the same solutions, and in particular the question of which of the two solutions is therefore the correct one. For this purpose one needs to take into account only the covariant components of the two conformal solutions, namely ${\hat{g}}_{\mu \nu }^{\left(C\right)}\left(s\right)=N\left(s\right){\hat{g}}_{\mu \nu }\left(r\left(s\right)\right)$ and ${\hat{g}}_{\mu \nu }^{\left(C_{1}\right)}\left(s\right)=\frac{1}{N(s)}{\hat{g}}_{\mu \nu }\left(r\left(s\right)\right)$, respectively; it follows that only the first one, namely ${\hat{g}}_{\mu \nu }^{\left(C\right)}\left(s\right)$, exhibits the correct asymptotic behavior, requiring for all μ,ν=0,3:(101)$$\lim_{s\to +\infty }{\hat{g}}_{\mu \nu }^{\left(C\right)}\left(s\right)<\infty .$$

### 6.2. “Conformal” Einstein Field Equations

It is immediate to prove that the conformal fields ${\hat{g}}^{\left(C\right)}\left(s\right)$ and ${\hat{g}}^{\left(C_{1}\right)}\left(s\right)$ satisfy corresponding, i.e., “*conformal*”, Einstein field equations. The form of such equations for the conformal fields ${\hat{g}}^{\left(C\right)}\left(s\right)$ and ${\hat{g}}^{\left(C_{1}\right)}\left(s\right)$ follows, in fact, in a straightforward way by direct comparison with Equation (Equation 31). Thus, for example, in the case of ${\hat{g}}^{\left(C\right)}\left(s\right)$ the corresponding realization of EFE takes the form
(102)$$-\sigma R_{\mu \nu }\left({\hat{g}}^{\left(C\right)}\left(s\right)\right)+\frac{\sigma }{2}{\hat{g}}_{\mu \nu }^{\left(C\right)}\left(s\right)R\left({\hat{g}}^{\left(C\right)}\left(s\right)\right)-\sigma {\hat{g}}_{\mu \nu }^{\left(C\right)}\left(s\right)\Lambda_{QM}\left({\hat{g}}^{\left(C\right)}\left(s\right)\right)=0.$$

The proof is immediate. In fact, one notices, first, that thanks to Equation (Equation 95) the Ricci tensor, just as the Riemann tensor, remains invariant. Second, thanks to Equations (Equation 87) and (Equation 97)
(103)$${\hat{g}}_{\mu \nu }\left(s\right)R\left(\hat{g}\left(s\right)\right)={\hat{g}}_{\mu \nu }^{\left(C\right)}\left(s\right)R\left({\hat{g}}^{\left(C\right)}\left(s\right)\right),$$
and, third, that similarly thanks to Equation (Equation 99) it follows
(104)$${\hat{g}}_{\mu \nu }\left(s\right)\Lambda_{QM}\left(\hat{g}\left(s\right)\right)={\hat{g}}_{\mu \nu }^{\left(C\right)}\left(s\right)\Lambda_{QM}\left({\hat{g}}^{\left(C\right)}\left(s\right)\right).$$

An equivalent proof of Equation (Equation 102) follows from Equation (Equation 39). In fact dividing it term by term by N(s) one obtains:(105)$$-\sigma {\hat{R}}_{\mu \nu }+\frac{\sigma }{2}{\hat{g}}_{\mu \nu }^{\left(d\right)}\left(s\right)\frac{R(\hat{g}\left(s\right))}{N(s)}-\sigma {\hat{g}}_{\mu \nu }^{\left(d\right)}\left(s\right)\frac{\Lambda_{QM}\left(\hat{g}\left(s\right)\right)}{N(s)}=0,$$
which, in the validity of Equations (Equation 96) and (Equation 98), recovers Equation (Equation 102) again.

### 6.3. “Conformal” Gaussian Quantum PDF and Quantum Continuity Equations

As a final issue, it should be mentioned that the conformal fields (Equation 87) and (Equation 88) are also consistent with:

(a) The prescription of the Gaussian quantum PDF $\rho (\Delta g-\hat{g}\left(s\right))$ defined by Equation (Equation 15) and with exponent (Equation 16).

(b) The quantum continuity Equation (Equation 128) (see Appendix B).

To prove that indeed Equations (Equation 87) and (Equation 88) represent admissible solutions it is sufficient to notice that the Gaussian PDF (Equation 15) remains unchanged under the transformation
(106)$$\left\{\begin{array}{c} \left[\Delta g_{{}^{\alpha \beta }}-{\hat{g}}_{{}^{\alpha \beta }}\left(s\right)\right]\to \left[\Delta g_{{}^{\alpha \beta }}-{\hat{g}}_{{}^{\alpha \beta }}\left(s\right)\right]N\left(s\right)\equiv \Delta g_{{}^{\alpha \beta }}^{\left(C\right)}-{\hat{g}}_{{}^{\alpha \beta }}^{\left(c\right)}\left(s\right),\\ \left[\Delta g^{\alpha \beta }-{\hat{g}}^{\alpha \beta }\left(s\right)\right]\to \left[\Delta g^{\alpha \beta }-{\hat{g}}^{\alpha \beta }\left(s\right)\right]\frac{1}{N(s)}\equiv \Delta g^{(C)\alpha \beta }-{\hat{g}}^{(c)\alpha \beta }\left(s\right), \end{array}\right.$$
with Ω(s)=N(s) or 1/N(s). Equation (Equation 106) are obtained invoking the transformations (Equation 87) and (Equation 91). This means that the Gaussian PDF (Equation 15) holds both for the stationary solution $\hat{g}\left(s\right)$ as well as for arbitrary simultaneous conformal solutions of the type (Equation 87)–(Equation 91). As a consequence it follows
(107)$$\rho \left(\Delta g-\hat{g}\left(s\right)\right)=\rho \left(\Delta^{\left(C\right)}g-{\hat{g}}^{\left(C\right)}\left(s\right)\right).$$

The equation of continuity (Equation 128) remains similarly invariant. The proof follows by noting that, upon denoting $V_{\mu \nu }^{\left(C\right)}=\frac{1}{N(s)}V_{\mu \nu }$, one obtains
(108)$$\frac{\partial }{\partial g_{\mu \nu }}\left(V_{\mu \nu }\rho \left(\Delta g-\hat{g}\left(s\right)\right)\right)=\frac{\partial }{\partial g_{\mu \nu }^{\left(C\right)}}\left(V_{\mu \nu }^{\left(C\right)}\rho \left(\Delta^{\left(C\right)}g-{\hat{g}}^{\left(C\right)}\left(s\right)\right)\right).$$

### 6.4. Conformal Fields as Possible New MFT

We conclude that:-Both ${\hat{g}}^{\left(C\right)}\left(s\right)$ and ${\hat{g}}^{\left(C_{1}\right)}\left(s\right)$ define conformal fields, which by construction satisfy the required orthogonality conditions.-Both fields fulfill suitable Einstein field equations, with suitably (scaled-down or increased) values of the Riemann 4−scalar and cosmological constant.-The prescription of the Gaussian quantum PDF remains unchanged.-The quantum continuity equation is fulfilled also in the case of conformal field.

This proves that in principle both fields ${\hat{g}}^{\left(C\right)}\left(s\right)$ and ${\hat{g}}^{\left(C_{1}\right)}\left(s\right)$ can be treated as MFT in place of the background MFT $\hat{g}\left(s\right)$. However, of the two conformal fields defined above, only the first one, ${\hat{g}}^{\left(C\right)}\left(s\right)$, which is constructed in terms of ${\hat{g}}_{\mu \nu }^{\left(C\right)}\left(s\right)={\hat{g}}_{\mu \nu }^{\left(d\right)}\left(s\right)$, is actually regular in the origin, in the sense that the regularity condition (Equation 76) holds. The two solutions nevertheless coincide when N(s)=1. Therefore, outside the EH of the BH, the two solutions might in principle coexist giving rise to different possible physical scenarios. This leaves us with the possible physical implications to be discussed in the next section.

## 7. Physical Interpretation

A general comment is in order about the physical interpretation of the mathematical scale-transformed solution pointed out above for the regularization of classical black hole singularities of space-time. The starting consideration concerns the fact that the quantum gravitational field possesses a manifestly covariant Hamiltonian dynamics, which is a direct consequence of the analogous Hamiltonian structure holding for classical GR and the adoption of a synchronous variational principle for the derivation of the Einstein field equations. The synchronous variational formulation in fact is characterized by adoption of superabundant variables and the distinction between the variational ($g_{\mu \nu }\left(s\right)$) and background (${\hat{g}}_{\mu \nu }\left(s\right)$) tensor fields, whereby the variational and background ones are allowed to carry different physical properties. More precisely, in such a picture the background metric tensor ${\hat{g}}_{\mu \nu }\left(s\right)$ has a geometrical connotation, in the sense that it is normalized so that ${\hat{g}}_{\mu \nu }\left(s\right){\hat{g}}^{\mu \nu }\left(s\right)=\delta_{\mu }^{\mu }$, it raises/lowers tensor indices and defines the Christoffel symbols and the background Ricci tensor. In addition, the same tensor ${\hat{g}}_{\mu \nu }$ is by definition characterized by having identically-vanishing covariant derivative, namely ${\hat{\nabla }}_{\alpha }{\hat{g}}_{\mu \nu }=0$. This equation defines the Christoffel symbols in terms of the metric tensor and is known in the literature as metric-compatibility condition. Borrowing a term from plasma ideal magnetohydrodynamics, we can interpret it as a “frozen-in” condition that establishes the link between the space-time geometrical structure and the gravitational tensor field, so that we can say that the metric field is the geometry. On the contrary, in the same framework, the field $g_{\mu \nu }\left(s\right)$ is allowed to have non-vanishing covariant derivative, i.e., ${\hat{\nabla }}_{\alpha }g_{\mu \nu }\left(s\right)\neq 0$, so that $g_{\mu \nu }\left(s\right)$ can acquire a non-null generalized kinetic energy. When canonical quantization is performed on the Hamiltonian structure, this generates a quantum gravitational field $g_{\mu \nu }\left(s\right)$ characterized by non-vanishing canonical momenta. The remarkable consequence is that, in the realm of quantum theory, while the background field ${\hat{g}}_{\mu \nu }\left(s\right)$ keeps on retaining its geometric meaning consistent with the picture of GR, the field $g_{\mu \nu }\left(s\right)$ acquires the physical meaning of a quantum field that is permitted to deviate from ${\hat{g}}_{\mu \nu }\left(r\right)$ and to exhibit a dynamics over the background space-time, thus violating at the quantum level the frozen-in condition ${\hat{\nabla }}_{\alpha }{\hat{g}}_{\mu \nu }\left(s\right)=0$ (which is nevertheless warranted at classical level, see details in Ref. [20]). This is precisely the feature that allows the quantum regularization of the classical singularity to be reached, as expressed by the scale-transformed solution reported above. In fact, the quantum gravitational field $g_{\mu \nu }\left(s\right)$ is no longer forced to follow the background geometry, but can deviate from it. Hence, while the classical metric tensor diverges with the geometry at the BH singularity, violation of the frozen-in condition for $g_{\mu \nu }\left(s\right)$ due to non-vanishing canonical momenta makes it possible to escape the BH singularity with a regular behavior. This feature is peculiar and unique of CQG-theory with respect to other quantum-gravity models proposed in the literature. In the present picture, manifest covariance is preserved and it is not the geometry to be quantized, but the field, while the background space-time metric tensor is obtained as a consistent solution of quantum-modified Einstein field equations.

Based on these conceptual preliminaries, the following physical scenarios can be distinguished in the framework of CQG-theory.

### 7.1. Inner BH Domain Behavior

CQG-theory allows for the existence of a *unique regular background MFT solution that holds in all singular BH solutions considered here*. Such a solution is realized by the conformal field ${\hat{g}}^{\left(C\right)}\left(s\right)$ in which the scale form factor N(s) tends to zero in the central position r=0, which corresponds to the limit
(109)$$\lim_{s\to \infty }N\left(s\right)=0,$$
and tends to unity on the EH (where $N(s_{o})=1$). As shown above, in the same limit the covariant and countervariant components ${\hat{g}}_{\mu \nu }^{\left(C\right)}\left(s\right)$ and ${\hat{g}}^{(C)\mu \nu }\left(s\right)$ remain finite. This explains also how CQG-theory actually cures the BH singularities and answers the first open question pointed out in the introduction (*preliminary issue #1*). One can envisage why this happens and what is the role of the cosmological constant. As shown elsewhere [23], an ubiquitous feature that occurs in the quantum-modified EFE is the appearance of a non-vanishing quantum cosmological constant. Such a cosmological constant enters in arbitrary MFT solutions of the same equation. However, as shown above (see Section 6) in the case of conformal MFT solutions the cosmological constant is modified in terms of the scale form factor N(s) or its reciprocal 1/N(s).

Let us now consider the possible connection with (asymptotic/localized) inflationary regimes (see also *preliminary issue #4*). Also in this case the answer is positive, in the sense that in the same limit the effective quantum cosmological constant $\Lambda_{QM}({\hat{g}}^{\left(C\right)}\left(s\right))$ diverges
(110)$$\lim_{s\to \infty }\Lambda_{QM}({\hat{g}}^{\left(C\right)}\left(s\right))=\lim_{s\to \infty }\frac{\Lambda_{QM}}{N(s)}=+\infty ,$$
which means that the conformal solution *becomes infinitely inflationary*. In other words, a characteristic feature of the occurrence of ${\hat{g}}^{\left(C\right)}\left(s\right)$ is necessarily its infinite-inflation property.

### 7.2. Intermediate Domain Behavior

As a work hypothesis we shall assume that the scale form factor N(s) is a continuous function across all EHs. This means, in particular, that in the internal EH necessarily one should expect *N* to be equal to unity. This implies that it should be $N(s_{o})$=1 also in the outer side of the same EH. As a consequence in the intermediate domain one expects the scale form factor N(s) to be either constant (N(s)=1) or monotonically increasing as a function of *s* (see Equation (Equation 79)). Then the basic implication is therefore that in such a domain two possible realizations exist for background MFT:The first one is provided by the conformal solution ${\hat{g}}^{\left(C\right)}\left(s\right)$. In such a case the corresponding effective cosmological constant is provided by Equation (Equation 99). In this case N(s) is a monotonically increasing function of *s* but is also bounded from above. This implies that the effective cosmological constant should *decrease toward the outer regions of the universe* (included in the domain inside the deSitter space-time) *but remain bounded from below*.The second possible realization is provided instead by the conformal solution ${\hat{g}}^{\left(C_{1}\right)}\left(s\right)$. In this case the corresponding effective cosmological constant is provided by Equation (Equation 100). Again for a monotonically increasing scale form factor N(s) this means that effective cosmological constant must *increase toward the outer regions of the universe* (which are inside the deSitter space-time) *but remain similarly bounded from above.* If N(s)≫1 then such a case corresponds to an *inflationary solution*, i.e., characterized by a strong enhancement of the effective cosmological constant for which $\Lambda_{QM}\left({\hat{g}}^{\left(C_{1}\right)}\left(s_{1}\right)\right)$ is larger (or even much larger) than $\Lambda_{QM}({\hat{g}}^{\left(C_{1}\right)}\left(r\left(s_{o}\right)\right)$.

### 7.3. Exterior BH Domain

In the semi-infinite external domain, ruling out a possible divergent behavior (which would generate a singular background MFT solution), the only admissible behavior of the scale form factor N(s) is the one which asymptotically behaves so that
(111)$$\lim_{s\to \infty }N\left(s\right)=N_{\max}.$$

The implication is that also in this case two possible realizations exist for the background MFT:The first one is provided by the conformal solution ${\hat{g}}^{\left(C\right)}\left(s\right)$. In such a case the corresponding effective cosmological constant is provided by Equation (Equation 99). The scale form factor N(s) is a monotonically decreasing function of *s*, which in the limit s→∞ satisfies Equation (Equation 111). Such a solution should be viewed as the continuation of the corresponding conformal solution that holds in the intermediate domain. This implies that the corresponding effective cosmological constant should grow monotonically, reaching at infinity a stationary finite value.The second possible realization is provided, instead, by the conformal solution ${\hat{g}}^{\left(C_{1}\right)}\left(s\right)$. Again this can be regarded as the continuation of the analogous solution holding in the intermediate domain. In this case the initial value of the corresponding effective cosmological constant (provided by Equation (Equation 100)) can be expected larger (or even much larger) than $\Lambda_{QM}$ (*inflationary initial state*), while for s→∞ it decays monotonically reaching again at infinity a stationary value not necessarily identical with the other one indicated above.

### 7.4. The Initial Conformal Deformation of Space-Time

The emerging physical interpretation in the context of CQG-theory is therefore that quantum regularization of singular BHs is unique, at least in the inner BH domain described above. But the question is why such a state should be the privileged one, somehow selected by nature among all possible singular solutions.

Let us try to provide a possible physical explanation. Thus, if one introduces for definiteness the Boltzmann–Shannon (B-S) entropy [24]
(112)$$S_{BS}\left(\rho (\Delta g,s)\right)=-\int_{U_{g}}d\left(\Delta g\left(s\right)\right)\rho \left(\Delta g,s\right)\ln\rho \left(\Delta g,s\right),$$
and the notion of quantum expectation value
(113)$$\langle A\rangle =\int_{U_{g}}d\left(\Delta g\left(s\right)\right)A\rho \left(\Delta g,s\right)$$
for an arbitrary summable function A=A(Δg,s), one expects/requires the same B-S entropy $S_{BS}\left(\rho (\Delta g,s)\right)$ to be maximal at some initial proper time $s_{o}$ which can be chosen to coincide with the initial proper time introduced in the first subsection of Section 5. Thus, invoking the Principle of Entropy Maximization (PEM [33,34]) implies suitably prescribing the initial values of the quantum expectation values $\langle 1\rangle =1$, $N\left(s_{o}\right)\langle \Delta g_{\mu \nu }\left(s_{o}\right)\rangle$, $\frac{1}{N(s_{o})}\langle \Delta g^{{}_{\mu \nu }}\left(s_{o}\right)\rangle$ and $\langle \Delta g{\left(s_{o}\right)}^{2}\rangle \equiv \langle \Delta g_{\mu \nu }\left(s_{o}\right)\Delta g^{\mu \nu }\left(s_{o}\right)\;\rangle$, i.e., more precisely setting: (114)$$\begin{array}{ccc} N\left(s_{o}\right)\langle \Delta g_{\mu \nu }\left(s_{o}\right)\rangle & = & {\hat{g}}_{\mu \nu }^{\left(C\right)}\left(s_{o}\right)\equiv N\left(s_{o}\right){\hat{g}}_{\mu \nu }\left(s_{o}\right), \end{array}$$(115)$$\begin{array}{ccc} \frac{1}{N(s_{o})}\langle \Delta g^{{}_{\mu \nu }}\left(s_{o}\right)\rangle & = & {\hat{g}}^{(C)\mu \nu }\left(s_{o}\right)\equiv \frac{1}{N(s_{o})}{\hat{g}}^{\mu \nu }\left(s_{o}\right), \end{array}$$(116)$$\begin{array}{ccc} \langle \Delta g{\left(s_{o}\right)}^{2}\rangle & = & 8r_{th}^{2}. \end{array}$$

Notice that here $N(s_{o})$, ${\hat{g}}_{\mu \nu }\left(s_{o}\right)$, ${\hat{g}}^{\mu \nu }\left(s_{o}\right)$ identify classical observables, while $r_{th}^{2}$ is a quantum parameter to be determined separately (for its evaluation see related discussion in Ref. [23]). Furthermore, $\Delta g_{\mu \nu }\left(s_{o}\right)$ and $\Delta g^{\mu \nu }\left(s_{o}\right)$ denote the covariant and countervariant components of the stochastic displacement tensor, respectively, while ${\hat{g}}_{\mu \nu }^{\left(C\right)}\left(s_{o}\right)$, ${\hat{g}}_{\mu \nu }\left(s_{o}\right)$ and ${\hat{g}}^{(C)\mu \nu }\left(s_{o}\right),{\hat{g}}^{\mu \nu }\left(s_{o}\right)$ are the components of the conformal tensor field ${\hat{g}}^{\left(C\right)}\left(s_{o}\right)$ (defined by Equation (Equation 87)) and of the background MFT $\hat{g}\left(s_{o}\right)$, respectively. It is then possible to show that PEM requires $\rho (\Delta g\left(s_{o}\right),s_{o})$ to be again a Gaussian PDF of the form:(117)$$\rho \left(\Delta g\left(s_{o}\right),s_{o}\right)=\rho_{G}\left(\Delta g\left(s_{o}\right)-\hat{g}\left(s_{o}\right)\right)\equiv \frac{1}{\langle \langle 1\rangle \rangle }\exp\left\{-\frac{{\left(\Delta g\left(s_{o}\right)-\hat{g}\left(s_{o}\right)\right)}^{2}}{r_{th}^{2}}\right\},$$
with $\langle \langle 1\rangle \rangle$ denoting here the normalization constant
(118)$$\langle \langle 1\rangle \rangle =\int_{U_{g}}d\left(\Delta g\left(s_{o}\right)\right)\exp\left\{-\frac{{\left(\Delta g\left(s_{o}\right)-\hat{g}\left(s_{o}\right)\right)}^{2}}{r_{th}^{2}}\right\}$$
(see also Appendix B), and ${\left(\Delta g\left(s_{o}\right)-\hat{g}\left(s_{o}\right)\right)}^{2}$ being defined according to Equation (Equation 106). The initial conditions (Equation 114)–(116), together with the normalization $\langle 1\rangle =1$, prescribe the initial values of $N(s_{o})$ and $\rho (\Delta g,s_{o})$. Validity of the quantum continuity Equation (Equation 128) then implies that ρ(Δg(s),s) is given according to Equation (Equation 131) in Appendix B. We notice that if PEM is required to hold for arbitrary $s\geq s_{o}$ then it follows necessarily that the condition (Equation 127) reported in Appendix B must apply. This means that the background metric tensor must be defined also in the limit s→∞. This permits in turn the same conformal tensor field ${\hat{g}}^{\left(C\right)}\left(s\right)$ to be everywhere regular (this condition to be intended again in the sense of the regularity conditions (Equation 68)). As a fundamental consequence the correspondence
(119)$$\hat{g}\left(s\right)\to {\hat{g}}^{\left(C\right)}\left(s\right)$$
effectively generates a *conformal deformation of space-time* whereby the differential-manifold structure of space time $\left\{Q^{4},\hat{g}\left(s\right)\right\}$ is replaced with $\left\{Q^{4},{\hat{g}}^{\left(C\right)}\left(s\right)\right\}$.

This provides an answer to the issue raised above: since PEM holds at arbitrary proper times, it must hold also asymptotically so that ${\hat{g}}^{\left(C\right)}\left(s\right)$ must exist also in the limit s→∞, thus ultimately requiring the background metric tensor, represented by ${\hat{g}}^{\left(C\right)}\left(s\right),$ to be regular in the same limit.

## 8. Conclusions

It is generally acknowledged that space-time singularities, particularly black hole (BH) ones, actually play a crucial role in general relativity (GR). In fact, apart their ubiquitous presence in the optically accessible (or non accessible) universe, the very existence of singular BHs represents a crucial conceptual issue. Indeed it is generally agreed that these singularities arise because of the failure of classical GR to describe them properly. The prevailing opinion in fact is that such singularities, which occur at the classical level, i.e., characterize the solutions of the Einstein field equations, are just the manifestation of possible but still unknown underlying quantum effects that arise in the presence of extremely intense gravitational fields. Thus, the proper understanding of the role of quantum gravity (QG) in this context becomes increasingly urgent and meaningful.

The issue in turn is intimately related to the kind of QG-theory required for such a task.

As shown here a convenient choice is represented by a theory that, by construction, should satisfy both the principles of general covariance and of manifest covariance with respect to the group of local point transformations (LPTs), i.e., coordinate diffeomorphisms mutually mapping in each other different GR frames. Our claim is that the manifestly covariant quantum gravity theory (CQG-theory) recently proposed fits well into the scheme. Indeed, the presence of extremely intense classical gravitational fields, with the implied consequence of possibly-related relativistic (or ultra-relativistic) particle effects, suggests that the gravitational field should be properly treated, a fact that rules out the adoption of possible approximations related to the (hitherto) classical structure of space-time. The obvious consequence is that such a theory should be set in manifestly covariant form with respect to arbitrary coordinate transformations which leave unchanged the structure of space-time. In other words it should always be possible to cast such a theory in explicit 4−tensor form with respect to the (yet to be determined) local space-time structure. In addition, one should agree that, in order for a quantum theory to be possible at all, it is obvious that a classical Hamiltonian structure should be (possibly non-uniquely) associated with the Einstein Field Equations. Needless to say, both the classical and corresponding quantum Hamiltonian structures should be manifestly covariant. These features are all embodied in CQG-theory.

In our view the adoption of such a type of tensor setting is actually expedient for the identification of the conformal background metric field tensors (MFT) described here. In this paper we have shown, in fact, that CQG-theory permits the explicit prescription of a suitable 4−scalar N(s), denoted as scale form factor, which allows their explicit determinations. Indeed, the prescription of N(s) as a function of the proper-time *s* (the arc length along a geodesic trajectory associated with the same background MFT) follows from a suitable set of manifestly covariant quantum Hamilton equations. Such equations, a chief characteristic of CQG-theory, depend on the geometry of space-time, i.e., the background Ricci tensor. In turn the latter depends on the quantum-produced cosmological constant, which in the context of CQG-theory, is found to be generated by the Bohm vacuum graviton interaction. This explains its ubiquitous nature and the fact the cosmological constant actually can affect also strong-field domains arising in the vicinities and particularly inside BHs.

With these considerations in mind, the starting point of the paper has been the investigation of the quantum Hamilton equations of CQG-theory. In particular we have first shown the existence of non-stationary scale-transformed solutions of the CQG-quantum Hamilton equations of the form ${\hat{g}}^{\left(d\right)}\left(s\right)=N\left(s\right)\hat{g}\left(s\right)$, with N(s) denoting a deterministic proper-time dependent scale-form factor and $\hat{g}\left(s\right)$ a particular solution of the quantum-modified Einstein field equations. We have shown that N(s) is uniquely determined by a suitable second-order ODE subject to prescribed initial conditions. The qualitative properties of the scale-form factor in different cases have been categorized distinguishing respectively: (1) an internal problem (inside the inner EH); (2) intermediate problem (between two EHs); (3) an external problem (outside the outer EH).

The main results concern:The discovery of a regular conformal representation of the background MFT that holds inside the BH domain in principle for arbitrary singular BH solutions. The regularization effects is purely quantum and arises due to the combined effect of the quantum-produced cosmological constant ($\Lambda_{QM}$) together with the Hamiltonian character of the underlying quantum hydrodynamic equations.The prediction of the large-scale behavior of the corresponding external conformal background MFTs, i.e., occurring in the external domains of the BH. Such predictions are obtained based on the assumption of continuity of the scale form factor N(s) across event horizons.

However, further notable features emerge which provide further physical insight regarding:How CQG-theory actually can cure BH singularities, giving rise to a suitable quantum-modified background metric field tensor (MFT).The role of the cosmological constant and how its quantum character actually affects the regularization of singular space-time solutions.The identification of the possible large-scale effects produced by the local quantum modifications of MFT.The possible connection between the occurrence/prediction of asymptotic/local inflationary regimes, characterized by high values of the cosmological constant and the expected phenomenon of BH-singularities-quenching.

These conclusions suggest a possible new mechanism of quantum-regularization of BHs, with profound physical implications on the large-scale structure of the universe. As shown here, in the context of CQG-theory, this is brought about by the occurrence of a conformal deformation of space-time whereby the singular space-time $\left\{Q^{4},\hat{g}\left(s\right)\right\}$ is replaced with $\left\{Q^{4},{\hat{g}}^{\left(C\right)}\left(s\right)\right\}$, which is generated by the regular conformal background MFT ${\hat{g}}^{\left(C\right)}\left(s\right)$.

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
