# Peer review of "The Quantum Regularization of Singular Black-Hole Solutions in Covariant Quantum Gravity"

_entropy, 2021, doi:10.3390/e23030370_

Round 1

Reviewer 1 Report

The authors present an interesting scenario where they propose a Quantum formalism of gravity in order to solve some of the problems observed at the classical level, as for example, singularity problems, the cosmological constant, among other problems. My concern is about the cosmological constant and how this formalism can really help to solve the 120 orders of magnitude problem or how can it derive a cosmological constant naturally. From one side, it seems that the factor $N(s)$ helps to reduce the cosmological constant value. But how can we justify Quantically $N(s)$? 

My second concern are the scales developed by the cosmological constant. It is known that when the CC and the Newtonian constant are both present, they develop a third scale which is a mix of the CC scale and the Newtonian scale represented through the gravitational radius GM. Could the authors explain if such scale might emerge naturally in this formalism or what is its meaning? Please take a look at Nuovo Cimento Vol. 91 B, N. 1, 11 Gennaio (1986); Class.Quant.Grav. 26 (2009) 125006; Phys. Rev. D 54
(1996), 6312-6322; Universe 3 (2017) 2, 45; Class. quant. Grav. 23, 485.

In addition, can the authors explain how the Hawking radiation can emerge in this scenario? Does it emerge in the same way as it is traditionally derived or should we expect some changes? See for example the original derivation of Hawking in Commun.Math.Phys. 43 (1975). 

Another point to consider is if your proposal converges or if it has similarities with modifications of gravity when it deals with the cosmological constant problem. There are cases for example when in some theories, the emergence of the cosmological constant can be perceived as a consequence of a gravitational sigma-model with the corresponding phase transitions. See for example EPL 115 (2016) 3, 31001; arXiv:1512.06838 [hep-th]; Universe 5 (2019) 166.

Finally, can the authors explain what can we expect about the propagation of gravitational waves inside their formulation? What changes appear? See for instance  Acta Phys.Polon.B 41 (2010) 911-925; Mod.Phys.Lett.A 28 (2013) 1350019; Phys. Rev. Lett. 121, 251103 (2018). It seems that the cosmological constant affects the propagation of gravitational waves in general. Besides that, the authors are proposing Quantum corrections to the singularities. Can these arrangements affect in some sense the propagation of gravitational waves?

If the authors could arrange the paper in agreement with these comments, the paper would look much better and it will call the attention of a wider audience. Additionally, in this way, the authors will find interesting open questions and possibly unexpected new phenomena. 

After the arrangements, I would happily revise the paper again.  

Author Response

see enclosed reply

Reviewer 2 Report

The paper reports on regularisation of black hole singularities stemming from the original CQG approach to quantum gravity developed by the authors. Since the CQG is advertised as Hamiltonian and covariant at the same time, the authors should clarify better their approach, possibly without just repeating the technical details from [14-26], because the present introductory material is excruciatingly confusing (if I may quote the incipit from their abstract).

In the usual canonical approach to general relativity (GR), time is introduced by foliating the spacetime manifold in Cauchy hypersurfaces, which gives the gauge symmetry of GR the form of the Bergmann-Komar group. Naturally, there is no time evolution of the metric in GR, because the metric is defined globally in the spacetime, and the usual notion of time evolution just means moving from one sheet of the foliation to the next. In fact, in the ADM decomposition, one considers the 3-metric on the chosen Cauchy hypersurface (which defines DeWitt's superspace) and this is how time reappears. Moreover, DeWitt's superspace of 3-metrics and the covariant superspace of Vilkovisky and DeWitt for the 4-metric, are defined so that gauge symmetries are properly mod out.

The metric in CQG is treated as a (a priori unconstrained?) "4-tensor" (by which I assume they mean a rank 2 symmetric tensor under general coordinate transformations) which depends on "the 4-position r=r(s) along a local field geodesic trajectory". What is a 4-position? Do they mean that "r" represent the coordinates for a point in the spacetime manifold? If so, why the fancy notation? More importantly, what is a field geodesic trajectory? A geodesic in any space is defined by a notion of parallel transport or by a metric, so what is the metric in this CQG field space? Is it related to the DeWitt metric in superspace? I do not find the Appendix clarifying at all in this respect. The crucial quantity "s" is further defined as "the Riemann length evaluated along a suitable family of field geodesics". Again this requires a metric in field space. Does that mean that given a spacetime point, one considers a trajectory in field space defined at that point parameterised by "s"? If so, what does that trajectory represent then for observers living in the spacetime? (A spacetime point is an event and events occur, they do not evolve). Although "s" refers to the evolution of the system in field apace, it is then called "proper time" in most of the paper. Is there a relation between "s" and the usual proper time in the ADM formalism (i.e. which observer measures that time)? Or does this notion of time in field space stem from a metric in field space whose nature is never really addressed?

Clarifying the meaning of "s" is particularly relevant, because all of the fundamental equations of CQG are about classical motions or quantum evolutions parametrised by "s". In fact, the remaining of the paper contains applications of the general formalism to the specific issue at hand. But if the foundations remain obscure, there is going to be little interest in trying to understand how the singularity is removed in CQG by some smart calculations.

Author Response

see enclosed file

Round 2

Reviewer 1 Report

The authors have addressed the comments and changes suggested by the referee. The paper can be published in the present form. 

Reviewer 2 Report

I think that the authors added enough introductory material to address my previous criticisms.